# Glutamine sensing licenses cholesterol synthesis

Bruna Martins Garcia [1], Philipp Melchinger [1], Tania Medeiros[1], Sebastian Hendrix [2], Kavan Prabhu[1], Mauro Corrado [3], Jenina Kingma [2], Andrej Gorbatenko[2], Soni Deshwal [1], Matteo Veronese[3], Luca Scorrano [4,5], Erika Pearce[6], Patrick Giavalisco [7], Noam Zelcer [2] & Lena Pernas [1,3,8,9 ✉]

## Abstract

The mevalonate pathway produces essential lipid metabolites such as cholesterol. Although this pathway is negatively regulated by metabolic intermediates, little is known of the metabolites that positively regulate its activity. We found that the amino acid glutamine is required to activate the mevalonate pathway. Glutamine starvation inhibited cholesterol synthesis and blocked transcription of the mevalonate pathway—even in the presence of glutamine derivatives such as ammonia and α-ketoglutarate. We pinpointed this glutamine-dependent effect to a loss in the ER-to-Golgi trafficking of SCAP that licenses the activation of SREBP2, the major transcriptional regulator of cholesterol synthesis. Both enforced Golgi-to-ER retro-translocation and the expression of a nuclear SREBP2 rescued mevalonate pathway activity during glutamine starvation. In a cell model of impaired mitochondrial respiration in which glutamine uptake is enhanced, SREBP2 activation and cellular cholesterol were increased. Thus, the mevalonate pathway senses and is activated by glutamine at a previously uncharacterized step, and the modulation of glutamine synthesis may be a strategy to regulate cholesterol levels in pathophysiological conditions.

**Keywords** HMGCR; Cholesterol; SREBP2; Nutrient Sensing; MFN2
**Subject Categories** Membranes & Trafficking; Metabolism

## Introduction

The mevalonate pathway is a pivotal node in cellular metabolism as it produces sterol and non-sterol isoprenoids that are essential for cellular function and viability (Goldstein and Brown, 1990). Cholesterol is the end product of mevalonate metabolism and a component of cellular membranes and myelin sheaths. It is also the precursor of signaling molecules such as steroid hormones that coordinate growth, development, and reproduction (Zhang and Liu,

2015). Because cholesterol is essential for organismal homeostasis but insoluble in aqueous environments, any imbalance in its levels can cause serious disease. A deficiency of cholesterol due to defects in genes required for its synthesis causes malformation syndromes (Porter and Herman, 2011). Conversely, the accumulation of cholesterol decreases membrane fluidity, disrupts membrane domains, and leads to inflammation, fibrosis, and nonalcoholic steatohepatitis (NASH) (Duewell et al, 2010; Ioannou, 2016; Song et al, 2021). Cells therefore face the challenge of maintaining mevalonate pathway activity to supply metabolites needed for growth while preventing the toxic accumulation of cholesterol.

To overcome this challenge, cells use sterol products of the mevalonate pathway to negatively regulate its activity at two critical nodes. The first of these nodes is the sterol regulatory element-binding protein 2 (SREBP2), an endoplasmic reticulum (ER) membrane-bound transcription factor that activates cholesterol synthesis (Brown and Goldstein, 1997). A decrease in ER-membrane cholesterol releases SREBP2 and the sterol sensor SCAP to which it is bound from the ER-resident anchor protein INSIG (Brown et al, 2018). The SCAP-SREBP2 complex is subsequently trafficked to the Golgi, where SREBP2 is proteolytically cleaved by the sequential action of site-1 protease (S1P; MBTPS1) and site-2 protease (S2P; MBTPS2) (Brown and Goldstein, 1997; Brown et al, 2018; DeBose-Boyd et al, 1999; Sakai et al, 1998). The processed form of SREBP2 enters the nucleus and induces the expression of cholesterol synthesis genes (Brown and Goldstein, 1997; Brown et al, 2018; DeBose-Boyd et al, 1999; Sakai et al, 1998). When cholesterol in ER membranes exceeds 5%, the binding of sterols to SCAP leads to the retention of the SCAP-SREBP2 complex by INSIG at the ER and thereby prevents SREBP2 activation (Brown et al, 2018). The second regulatory node is the ER-localized 3-hydroxy-3-methyl glutaryl coenzyme A reductase (HMGCR), the rate-limiting enzyme of the mevalonate pathway and the target of cholesterol-lowering statins (Schumacher and DeBose-Boyd, 2021). The sensing of oxysterols such as 25-hydroxycholesterol (25-HC) or lanosterol at the ER membrane promotes the extraction and proteasomal degradation of HMGCR (Schumacher and DeBose-Boyd, 2021). When sterols are depleted however, HMGCR is protected from degradation (Espenshade et al, 2002; Schumacher and DeBose-Boyd, 2021; Yang et al, 2002).

[1]Max Planck Institute for Biology of Ageing, Cologne, Germany. [2]Department of Medical Biochemistry, Amsterdam UMC, Amsterdam Institutes of Cardiovascular Sciences, and Gastroenterology Endocrinology and Metabolism, University of Amsterdam, Amsterdam, the Netherlands. [3]Cologne Excellence Cluster on Cellular Stress Responses in Aging-Associated Diseases (CECAD), University of Cologne, Cologne, Germany. [4]Department of Biology, University of Padova, Padova, Italy. [5]Venetian Institute of Molecular Medicine, Padova, Italy. [6]Bloomberg-Kimmel Institute for Cancer Immunotherapy and Department of Oncology, Johns Hopkins University School of Medicine, Baltimore, MD, USA. [7]Metabolomics Core Facility, Max Planck Institute for Biology of Ageing, Cologne, Germany. [8]Department of Microbiology, Immunology and Molecular Genetics, UCLA, Los Angeles, USA. [9]Howard Hughes Medical Institute, Chevy Chase, MD, USA. ✉E-mail: lfpernas@mednet.ucla.edu

The synthesis of cholesterol consumes over 100 units of ATP and requires 18 molecules of acetyl coenzyme A (acetyl-CoA), which is generated from mitochondrial citrate (Coleman and Parlo, 2021). Thus, an additional challenge cells face is tuning the activity of the mevalonate pathway to the availability of its inputs. One strategy by which cells do so in nutrient-poor conditions is through AMP-activated protein kinase (AMPK), a sensor of cellular energy that is activated by a low ATP/AMP ratio and inhibits cholesterol synthesis through the phosphorylation of HMGCR (Clarke and Hardie, 1990; Herzig and Shaw, 2018). During nutrient-replete conditions, the mevalonate pathway is positively regulated by the mechanistic target of rapamycin complex 1 (mTORC1), a protein kinase complex that controls anabolic processes and cell growth (Duvel et al, 2010; Liu and Sabatini, 2020). Whether other mechanisms exist to connect the mevalonate pathway to its metabolic inputs is unknown.

Here, we further investigate cholesterol metabolism and ask: to what extent is the activity of the mevalonate pathway governed by its precursors? To do so, we deprived cells of glucose and glutamine, the most important carbon sources of the citrate that fuels cholesterol synthesis (Yang et al, 2014). We found that limiting glutamine, but not glucose, inhibited cholesterol synthesis and transcriptionally repressed the mevalonate pathway. The effect of glutamine depletion was independent of glutamine anaplerosis, mTORC1 activity, global effects on translation and transcription, and known derivatives of glutamine. Of note, we found that ammonia, which was previously suggested to induce SREBP2, activated the mevalonate pathway in a manner dependent on its use for glutamine synthesis (Cheng et al, 2022). Glutamine was required for the transport-dependent activation of SREBP2 independently of INSIG, which retains SREBP2 at the ER through SCAP (Brown et al, 2018). In line with these results, enhanced uptake of glutamine in a model of mitochondrial dysfunction increased HMGCR abundance, SREBP2 activity, and cellular cholesterol. Our results identify glutamine as a regulatory input into the cholesterol pathway and suggest that glutamine sensing acts as an anabolic toggle-switch that couples precursor availability to sterol biosynthesis.

# Results

## Glutamine starvation inhibits cholesterol synthesis

To further understand the metabolic inputs that regulate the mevalonate pathway, we used mass spectrometry to measure cholesterol in cells cultured in media lacking either glucose or glutamine, the main nutrients that cells use to generate citrate. Following its synthesis in the mitochondria, citrate is transported to the cytosol for use in cholesterol synthesis (Fig. 1A) (Martinez-Reyes and Chandel, 2020). We found that cells cultured with glucose had slightly decreased levels of cholesterol relative to cells cultured with only glutamine for 8 h—despite having ~20% more total citrate (Fig. 1B–E). This result was unexpected given that cholesterol synthesis depends on the generation of acetyl-CoA from citrate, and raised the question of how glutamine deprivation reduced cholesterol levels.

In many cancer cells, citrate derived from glucose is exported from mitochondria for use in synthetic reactions while glutamine

fuels anaplerotic reactions that replenish intermediates of the TCA cycle (Fig. 1F; left) (Kornberg, 1965; Owen et al, 2002). We thus considered whether in the absence of glutamine, glucose-derived citrate was oxidized to sustain the TCA cycle, rather than exported into the cytosol for use in cholesterol synthesis (Fig. 1F; right) (DeBerardinis et al, 2007). To address this possibility, we examined the incorporation of $^{13}C$ from uniformly labeled $^{13}C$-glucose into citrate and cholesterol in cells cultured with or without glutamine. To ensure activation of the mevalonate pathway, cells were cultured in the absence of serum for all conditions, and thus exhibited synchronous cell cycle profiles (Appendix Fig. S1A,B). In glutamine-fed (gln+) cells, the majority of citrate had 2 labeled carbons (m + 2), the expected pattern following its generation in and export from mitochondria (Fig. 1F,G) (Owen et al, 2002; Yang et al, 2014). Glutamine-starved (gln-) cells however had a > 60% decrease in m + 2 citrate, while m + 4 and m + 6 citrate were increased by ~5- and ~100-fold, respectively (Fig. 1G). A similar trend was observed for the TCA cycle intermediates malate and succinate (Supp Fig. 1C). To test whether decreased anaplerosis drove the observed shift in citrate isotopologues, we supplemented gln- cells with α-ketoglutarate (αKG), the main anaplerotic derivative of glutamine (Kornberg, 1965; Owen et al, 2002). The addition of αKG was sufficient to restore m + 2 and m + 6 citrate pools to levels found in gln+ cells (Fig. 1G). It did not, however, rescue the >80% decrease in cholesterol synthesis in gln- cells as measured by $^{13}C$-incorporation into cholesterol (Fig. 1H). To exclude the possibility that glutamine-starvation decreased cholesterol synthesis through an indirect effect on glucose metabolism, we next incubated cells with $^{14}C$-acetate, which is incorporated into cholesterol following its conversion to acetyl-CoA in the cytosol (Coleman and Parlo, 2021). Consistent with our results using $^{13}C$-glucose, we found that glutamine-deprivation significantly reduced $^{14}C$-acetate incorporation into cholesterol, as did treatment with the HMGCR inhibitor simvastatin (Appendix Fig. S1D). Thus, glutamine is required for cholesterol synthesis in glucose-fed cells independently of its role in anaplerosis.

## Glutamine starvation inhibits the mevalonate pathway

How does glutamine starvation inhibit cholesterol synthesis? To address this question, we monitored levels of HMGCR because it catalyzes the first committed step of the mevalonate pathway (Schumacher and DeBose-Boyd, 2021). HMGCR was decreased as early as 2 h after glutamine withdrawal and was undetectable by 16 and 24 h in U2OS cells (Fig. 2A). Similar results were obtained for HeLa cells and primary human foreskin fibroblasts (HFFs) cultured without glutamine for 24 h (Fig. 2B,C). Thus, glutamine is required to maintain HMGCR in various human cell lines. Because HMGCR is the rate-limiting enzyme in cholesterol synthesis and its loss paralleled the decrease in cholesterol synthesis (Fig. 1H), we hereafter use HMGCR as a marker for mevalonate pathway activity (Schumacher and DeBose-Boyd, 2021).

## Glutamine synthesis from ammonia and glutamate sustains HMGCR in a cell-specific manner

Glutamine is classified as a non-essential amino acid because it can be synthesized from ammonia and glutamate by glutamine synthetase (GLUL) (Fig. 2D) (Zhang et al, 2017). This led us to ask why

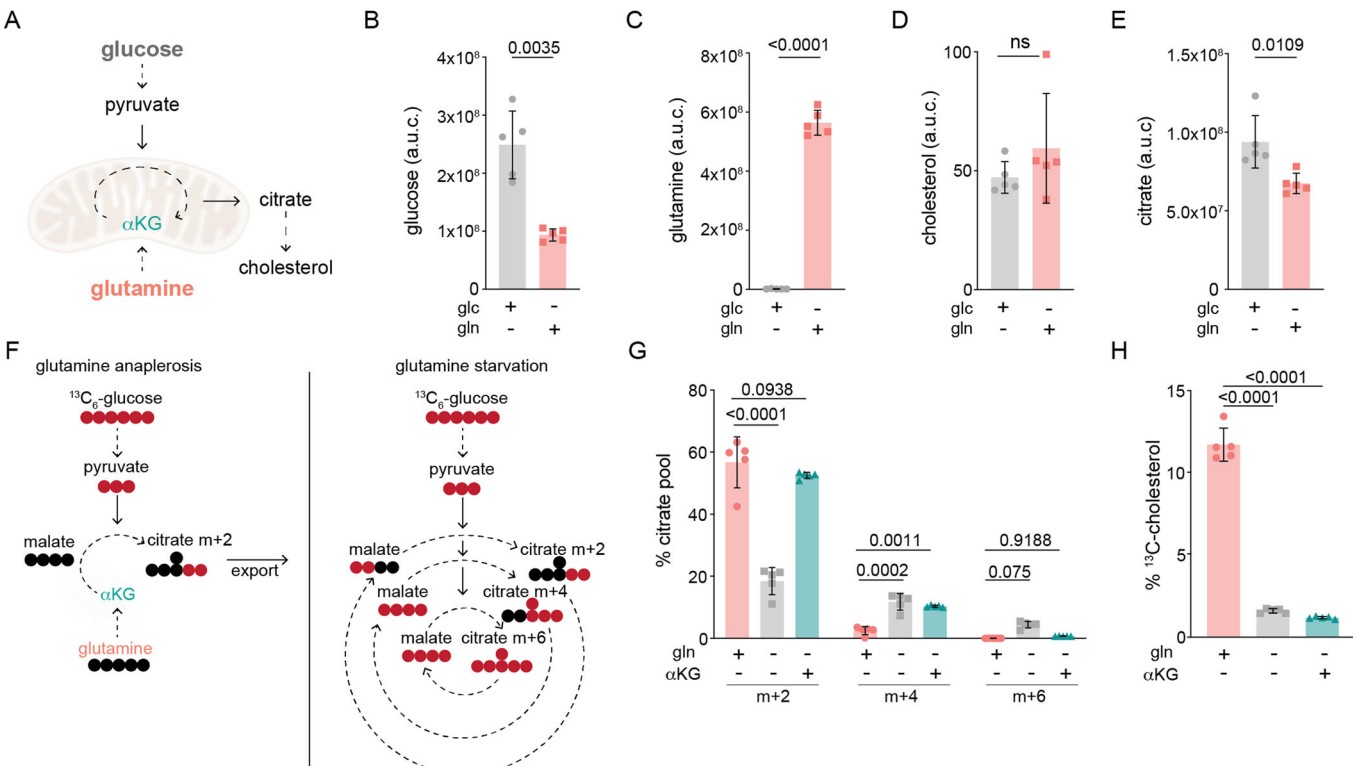

**Figure 1. Glutamine is required for cholesterol synthesis.**

(A) Main nutrients used to generate the citrate used for cholesterol synthesis are glucose via oxidation of pyruvate, and glutamine via α-ketoglutarate (αKG); dotted arrow represents 1+ steps. The abundance of total (B) glucose, (C) glutamine, (D) cholesterol, and (E) citrate in U2OS cells cultured with only glucose (glc) or only glutamine (gln) for 8 h. Data are mean ± s.d. of $n = 5$ independent cultures, Glc vs. Gln by unpaired t-test; a.u.c.: area under the curve. (F) Left: Simplified model of efflux of m + 2 citrate derived from $^{13}C_6$-glucose when anaplerosis is supplied by glutamine-derived αKG; right: oxidation of glucose feeds the TCA cycle in the absence of glutamine. Red circles represent $^{13}C$-carbons and black circles represent $^{12}C$-carbons; dotted arrow represents >1 reactions. (G) Mass isotopologues of citrate in U2OS cells cultured for 24 h with 25 mM $^{13}C_6$-glucose in the presence of 2 mM glutamine (gln+) or 1 mM αKG as indicated. Data are mean ± s.d. of $n = 5$ independent cultures; ns: not significant by two-way ANOVA. (H) Percentage of $^{13}C$-cholesterol in U2OS cells treated as in (G). Data are mean ± s.d. of $n = 5$ independent cultures; by one-way ANOVA. Cells were cultured in lipid-free media for all experiments.

endogenously synthesized glutamine did not maintain HMGCR levels in U2OS cells—at least minimally—during glutamine deprivation (Fig. 2A). To address this question, we examined GLUL RNAseq and proteomic quantitative profiling of cancer cell lines available through the Cancer Genome Atlas (Dataset EV1)(https://depmap.org/portal/ccle/). Of the 347 cell lines analyzed, U2OS cells were of the 10 with the lowest GLUL expression, and of the 10% with the lowest *GLUL* transcripts (Fig. 2E; Dataset EV1). We therefore hypothesized that U2OS cells lost HMGCR levels during glutamine starvation because they were deficient for GLUL activity. To test this hypothesis, we measured their incorporation of $^{15}N$-ammonia ($^{15}NH_3$) into glutamine. For comparison, we chose liver cancer cells (HepG2s) that opposite to U2OS cells were amongst the highest 10% of GLUL-expressing cell lines (Fig. 2E). After 24 h of glutamine starvation, $^{15}N$-labeled glutamine was less than 2% of total glutamine in U2OS cells and did not affect total glutamine and cholesterol pools (Fig. 2F–H). By contrast, in HepG2 cells >60% of glutamine was $^{15}N$-labeled, and the total glutamine and cholesterol pool were increased by >2-fold and 1.5-fold, respectively (Fig. 2F–H).

Unlike U2OSs, cell lines including HepG2, U87, LN229, and MDA468 that were previously used to demonstrate a role for ammonia in regulating lipid synthesis, are moderate to high

expressors of GLUL (Fig. 2E) (Cheng et al, 2022). This observation hinted that the effects of ammonia on HMGCR were dependent on glutamine synthesis, rather than ammonia-based signaling as suggested (Cheng et al, 2022). To address this possibility, we compared the effect of ammonia supplementation on HMGCR protein in HepG2 and U2OS cells cultured without glutamine. The addition of ammonia led to a robust induction of HMGCR in HepG2 cells, but had no effect in U2OS cells (Fig. 2I,J). We next tested whether forced expression of *GLUL* rendered U2OS cells similarly responsive to ammonia as HepG2 cells. We found that ammonia only induced HMGCR levels in U2OS cells expressing GLUL, unlike glutamine that induced HMGCR levels irrespective of GLUL expression (Fig. 2K). Thus, glutamine synthesis is required for ammonia to induce HMGCR levels in U2OS cells.

To next test whether glutamine synthesis was required for the induction of HMGCR by ammonia in HepG2 cells, we used L-methionine sulfoximine (MSX), a selective and irreversible inhibitor of GLUL (Rowe and Meister, 1970). We found that MSX treatment of HepG2 cells prevented the ammonia-induced increase in HMGCR (Fig. 2L). MSX treatment also reduced levels of HMGCR independently of ammonia and glutamine supplementation, pointing towards a contribution of endogenous glutamine synthesis to the

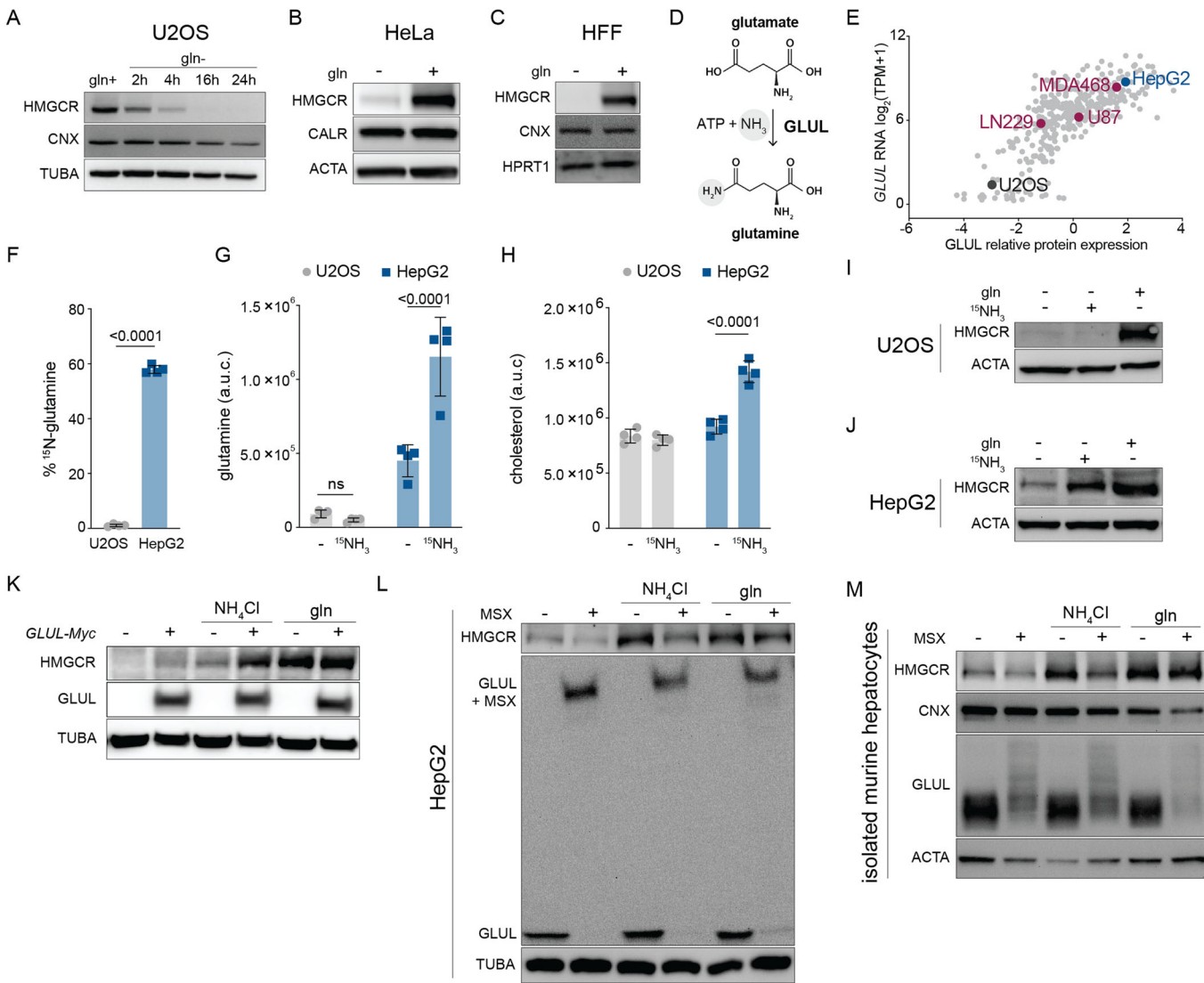

**Figure 2. Glutamine starvation inhibits the mevalonate pathway.**

(A) U2OS cells were cultured ± glutamine (gln+, gln-) for the indicated times and analyzed by immunoblotting for HMGCR, calnexin (CNX), and tubulin (TUBA). (B) HeLa cells and (C) primary human foreskin fibroblasts (HFFs) were cultured w/ or w/o glutamine (gln+, gln-) for 24 h and analyzed by immunoblotting for HMGCR, calreticulin (CALR), actin (ACTA) CNX, or hypoxanthine phosphoribosyl transferase (HPRT1) as indicated. (D) Glutamine synthetase (GLUL) generates gln from ammonia and glutamate. (E) GLUL relative protein expression and RNA $\log_2$ TPM + 1 in 347 cancer cell lines; data were obtained from previously generated data available at https://depmap.org/portal/ccle/. (F) Percentage of $^{15}$N-glutamine in U2OS and HepG2 cells cultured for 24 h w/o gln and supplemented for 24 h with 10 mM $^{15}$NH$_4$Cl. Data are mean ± s.d. of $n = 4$ independent cultures, analysis by unpaired t-test. Total pool size of (G) glutamine and (H) cholesterol from samples treated as in (F); data are mean ± s.d. of $n = 4$ independent cultures, analysis by two-way ANOVA. (I) U2OS and (J) HepG2 cells were starved of glutamine for 24 h, then treated for 24 h w/ gln or w/o gln ± 10 mM $^{15}$NH$_4$Cl. Samples were harvested and analyzed by immunoblotting for HMGCR and actin (ACTA). (K) U2OS cells expressing cDNA encoding for MYC-tagged GLUL were cultured with or without glutamine and 10 mM NH$_4$Cl for 24 h. Samples were analyzed by immunoblotting for HMGCR, GLUL, TUBA. Following glutamine starvation for 24 h, (L) HepG2s and (M) isolated murine hepatocytes were treated as indicated for 8 and 24 h, respectively, and analyzed by immunoblotting for HMGCR, GLUL, TUBA, CNX, and ACTA as indicated. Concentrations used: Gln 2 mM, 10 mM NH$_4$Cl, 500 µM methionine sulfoximine (MSX). Cells were cultured in lipid-free media for all experiments. Source data are available online for this figure.

regulation of HMGCR (Fig. 2L). Interestingly, we observed that MSX affected the migration pattern of GLUL, supporting its binding to and inhibition of GLUL in HepG2 cells (Fig. 2L). To test whether GLUL activity contributed to HMGCR in a physiologically relevant cell type, we tested the effects of MSX and ammonia in isolated primary murine hepatocytes. Ammonia induced HMGCR in a GLUL-dependent manner in primary hepatocytes, in line with the results obtained in HepG2 cells (Fig. 2L,M). Thus, glutamine synthesis

regulates cholesterol metabolism in a physiologically relevant and cell-type-specific manner.

## Glutamine, but not its derivatives, is required to maintain HMGCR expression

How does glutamine affect HMGCR levels? We first considered the possibility that glutamine promoted HMGCR stability. To address

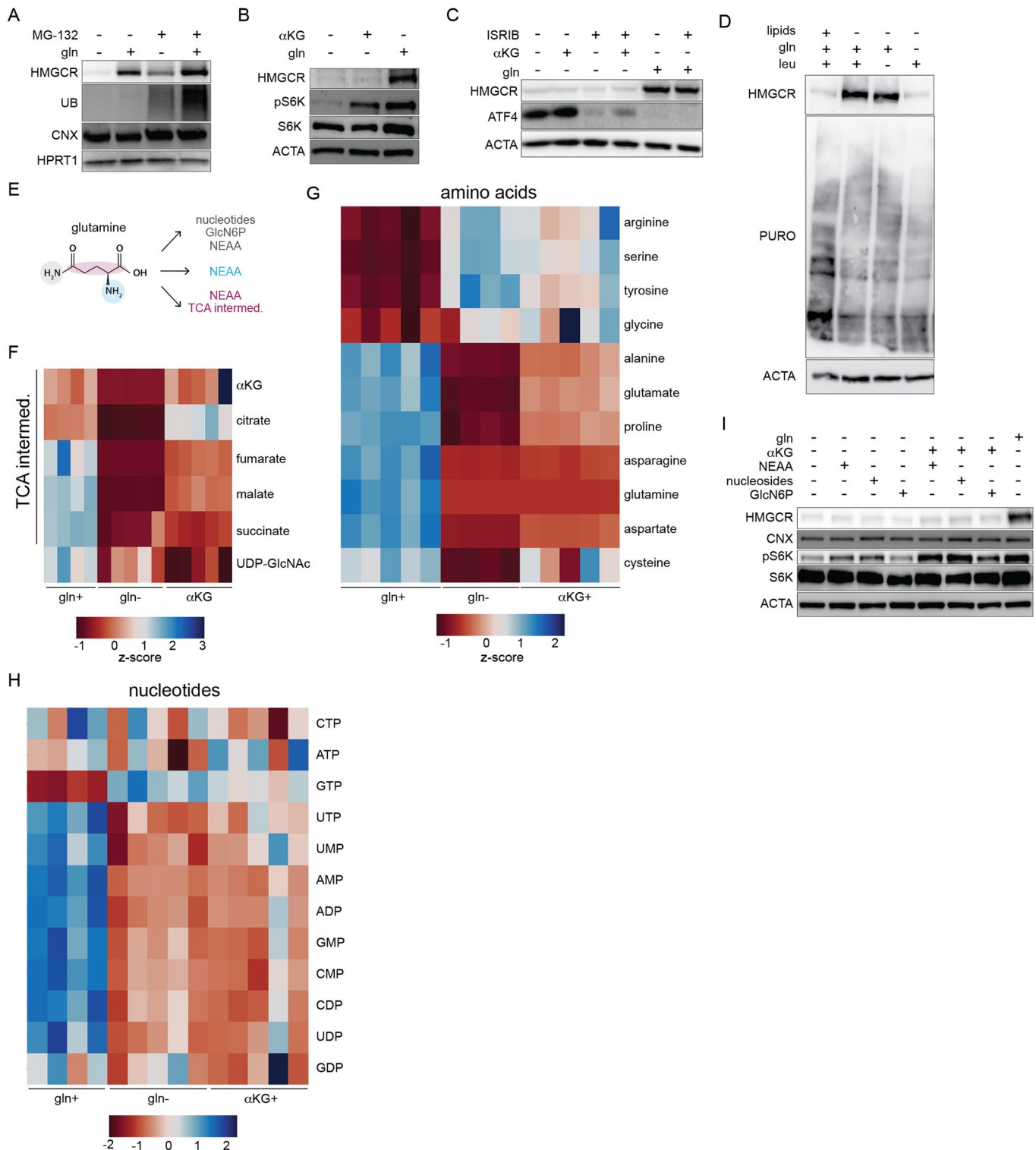

this question, we asked whether increased glutamine levels prevented sterol-induced degradation of HMGCR. However, culturing cells with 8 mM instead of 2 mM of glutamine did not prevent 25HC-induced degradation (Appendix Fig. S2A). Because HMGCR is regulated by ER-associated degradation (ERAD), we next asked if glutamine starvation triggered the ubiquitin/

proteasome system (UPS) (Schumacher and DeBose-Boyd, 2021). To do so, we treated cells with either the proteasomal inhibitor MG-132 or simvastatin, which enhances the expression of *HMGCR* and concomitantly blocks its ERAD. Neither treatment prevented the loss of HMGCR following glutamine withdrawal (Fig. 3A; Appendix Fig. S2B) (Schumacher and DeBose-Boyd, 2021).

**Figure 3.  Glutamine, but not its derivatives, is required for HMGCR expression.**

Glutamine-fed U2OS cells were cultured (**A**) ± glutamine (gln+, gln-) and ± MG-132 (10 µM) for 8 h; (**B**) for 24 h ± gln and ± αKG as indicated (**C**) ± gln and ± ISRIB (200 nM) ± αKG as indicated for 8 h. (**A–C**) Samples were analyzed by immunoblotting for HMGCR, calnexin (CNX), hypoxanthine phosphoribosyltransferase 1 (HPRT1), ubiquitin (UB), p70 S6 kinase (S6K), p70 phospho-S6K (pS6K), ATF4; and actin (ACTA). (**D**) U2OS cells were treated for as indicated for 8 h, at which point new protein synthesis was assayed by puromycin incorporation (lipids: 10% FBS; leu: 0.8 mM; gln, 2 mM). (**E**) Glutamine supplies carbons and nitrogen for the biosynthesis of TCA metabolites, glucosamine-6-phosphate (GlcN6P) that is used for UDP-GlcNAc synthesis, non-essential amino acids (NEAA), and nucleotides. Heat map of z-score normalization of the total pool size of (**F**) TCA-cycle metabolites and UDP-GlcNAc, (**G**) amino acids (AA), and (**H**) nucleotides in U2OS cells cultured for 24 h with 25 mM $^{13}C_6$-glucose in gln+ or gln- ± αKG (gln-; αKG+, respectively) conditions. Data are mean ± s.d. of $n = 5$ independent cultures. (**I**) U2OS cells were cultured w/o gln for 8 h w/ indicated supplements and concentrations; gln: 2 mM; αKG: 1 mM; NEAA: 100 µM (glycine, alanine, asparagine, aspartate, glutamate, proline and serine), nucleosides (cytidine 7.3 mg/L, guanosine 8.5 mg/L, uridine 7.3 mg/L, adenosine 8 mg/L, and thymidine 2.4 mg/L), and glucosamine (precursor to UDP-GlcNAc): 1 mM. Samples were analyzed by immunoblotting for HMGCR, CNX, pS6K, and ACTA. For all experiments, glutamine was used at 2 mM, αKG at 1 mM. Cells were cultured in lipid-free media for all experiments. Source data are available online for this figure.

Furthermore, our result that the levels of the ER chaperones calnexin and calreticulin were unaffected argued against an autophagy-mediated loss of HMGCR (Figs. 2A–C and 3A).

During glutamine starvation, mTORC1 is inactive and the integrated stress response (ISR) is induced, both of which lead to a reduction in global protein synthesis (Chen et al, 2014; Duran et al, 2012; Duvel et al, 2010; Sidrauski et al, 2013). Glutamine depletion could therefore indirectly reduce HMGCR levels through mTORC1 inhibition or the ISR. To test this possibility, gln- cells were treated with αKG, which maintains mTORC1 activity during glutamine starvation, and the ISR-inhibitor ISRIB (Duran et al, 2012; Sidrauski et al, 2013). Although αKG restored mTORC1 phosphorylation of its target ribosomal protein S6 kinase 1 (S6K1) and ISRIB blocked the activation of the key ISR effector ATF4, neither treatment prevented the loss of HMGCR during glutamine withdrawal (Fig. 3B,C). Consistent with these results, we found that bulk translation in U2OS cells as measured by puromycin incorporation was unaffected by glutamine starvation (Fig. 3D). Thus, glutamine starvation inhibits the mevalonate pathway independently of mTORC1 activity and the ISR.

Having excluded stress signaling-based effects of glutamine deprivation, we next asked if glutamine regulated the mevalonate pathway through a metabolic intermediate. Because glutamine-derived carbon and nitrogen fuel several biosynthetic reactions, we examined the set of metabolites that were depleted following glutamine withdrawal (Fig. 3E) (Zhang et al, 2017). As expected, metabolites that rely on glutamine for their synthesis including TCA cycle products, non-essential amino acids (NEAAs), nucleotides, and uridine diphosphate N-acetylglucosamine (UDP-GlcNAc) were markedly decreased after 24 h (Fig. 3F–H). We first tested the link between TCA intermediate levels and HMGCR loss. Supplementation with sodium pyruvate which feeds the TCA did not rescue HMGCR levels during glutamine starvation (Appendix Fig. S3A). Neither did the addition of αKG, the main anaplerotic derivative of glutamine, despite that it mitigated the loss of TCA metabolites during glutamine deprivation (Fig. 3B,F). In further support that glutamine insufficiency is not communicated through TCA cycle intermediates, we found that U2OS cells deficient for either citrate synthase (CS) or the mitochondrial citrate transporter (SLC25A1)—and which had ~90% lower citrate levels than those of wild-type (WT) cells—maintained HMGCR at levels higher than WT U2OS cells and similarly responded to glutamine deprivation (Appendix Fig. S3B–F).

We next tested the effect of NEAAs, nucleosides, and GlcNAc. None rescued HMGCR levels during glutamine starvation, even in

the presence of mTORC1-activating αKG (Fig. 3I). In line with these findings, the inhibition of glutaminolysis into glutamate and ammonia by BPTES did not reduce HMGCR in glutamine-replete conditions (Appendix Fig. S3G) (Robinson et al, 2007; Willis and Seegmiller, 1977). These results were consistent with our finding that ammonia sustained HMGCR in a manner dependent on glutamine synthesis (Fig. 2K,L). Furthermore, neither DON (6-diazo-5-oxo-L-norleucine), a reactive analog of glutamine, nor D-glutamine rescued levels of HMGCR during glutamine starvation (Appendix Fig. S3H,I). Thus, glutamine, but not its known intermediates, is sensed by the mevalonate pathway.

## Glutamine starvation transcriptionally represses cholesterol synthesis

Having established that glutamine positively regulated the mevalonate pathway, we next asked at what step does this occur? Our results that glutamine starvation drove the loss of HMGCR in a UPS-independent manner, despite treatment with simvastatin argued against its post-translational regulation (Fig. 3A; Appendix Fig. S2B). Furthermore, HMGCR expression was restored after 6+ hours of glutamine refeeding (Fig. 4A). Because prolonged exposure (72 h) to amino acid starvation leads to the repression of nascent RNA synthesis, we asked whether glutamine-starvation for 8 h, a time-point at which we observed a drastic reduction in HMGCR levels, compromised global transcription (Fig. 3D) (Pavlova et al, 2020). However, we found that labeled uridine was incorporated into nascent RNAs at similar levels in cells cultured with and without glutamine (Fig. 4B). We therefore next compared the transcriptome of cells cultured with (gln+) or without glutamine (gln-) for 8 h. Cells treated with αKG but not glutamine (gln- αKG+) were also analyzed to control for transcriptional changes that resulted from mTORC1 inhibition (Duran et al, 2012) (Dataset EV2). To identify glutamine-regulated genes, we applied two criteria using a false discovery rate (FDR) of <0.05: (1) difference between gln+ vs. gln- treatments (gln- and gln-αKG+ were grouped together) and; (2) no difference between gln- vs. gln-αKG+ (Fig. 4C). Analysis of gene ontology (GO) terms of the top 5% of changed genes revealed the global repression of genes in sterol biosynthesis-related pathways in gln- and gln-αKG+ cells (Fig. 4D). Among the top 25 glutamine-regulated mRNAs were those encoding for key enzymes in the cholesterol biosynthesis pathway including *HMGCR, SQLE*, and *FDFT1*, which we confirmed by quantitative PCR analysis (Fig. 4E; Appendix Fig. S4A–C). The result that this subset of mRNAs

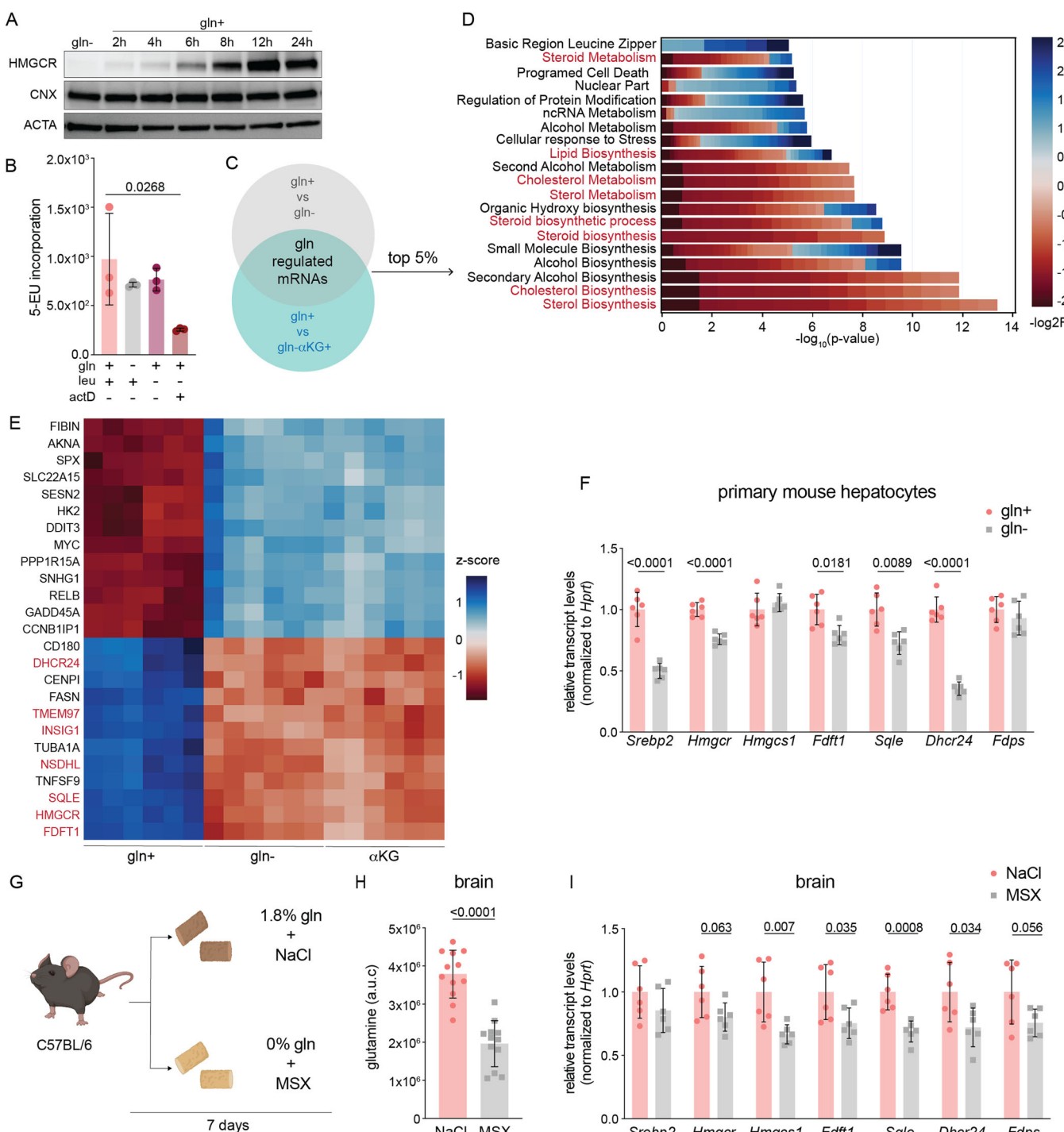

was also repressed in murine mammary carcinomas and primary murine hepatocytes cultured without glutamine led us to next ask whether glutamine regulated mevalonate pathway activity in vivo (Fig. 4F). To address this question, we examined mevalonate pathway transcript levels in brain and liver tissue—the main sites of in vivo cholesterol biosynthesis—of mice fed a control or glutamine-free diet (Fig. 4G)(Table EV1) (Zhang and Liu, 2015). The latter group was also treated with MSX at 48 h intervals to inhibit glutamine synthesis in vivo (Fig. 4G). Dietary and pharmacological glutamine deprivation led to a 50% reduction in total glutamine in brain tissue by day 7 and a corresponding decrease in mevalonate pathway transcripts (Fig. 4H,I). By contrast, neither glutamine levels nor mevalonate pathway transcripts were changed in liver tissue, suggesting the activation of pathways that sustain hepatic glutamine levels (Appendix Fig. S4D,E). Thus, glutamine deprivation transcriptionally represses cholesterol synthesis in cancer and primary cell lines, and in brain tissue in vivo.

**Figure 4. Glutamine starvation transcriptionally represses cholesterol.**

(A) U2OS cells were cultured with ±2 mM glutamine (gln) for the indicated time. Samples were analyzed by immunoblotting for HMGCR, calnexin (CNX), and actin (ACTA). (B) U2OS cells were treated as indicated for 8 h and nascent RNA synthesis was measured by 5-ethynyl-uridine (5-EU) incorporation; leu: 0.8 mM; gln, 2 mM; actinomycin D (actD) added 30 min prior to harvesting at 4 µg/mL. Data are shown as mean ± s.d. of $n = 3$ biological replicates by one-way ANOVA. (C) U2OS cells were cultured ±gln and ±αKG as indicated for 8 h and analyzed by RNAseq analysis. Differential expression was performed between "Control" and "Treatment" samples using limma/3.54.0 (Love et al, 2014). Gln+ samples were classified as Control whereas gln- samples and aKG+ samples were classified as Treatment (Sherman et al, 2022). (D) Pathway enrichment analysis was performed using the DAVID API on the top 5% of glutamine-responsive genes from experiment schematicized in (C); cholesterol-related processes are highlighted in red. (E) Heat map of z-score of expression values of the top 25 glutamine-regulated genes from the subset as described in (C); genes in red are SREBP2 targets. (F) mRNA levels in isolated primary hepatocytes that were 24h-starved of gln and treated with 500 mM methionine sulfoximine (MSX) ± 2 mM of gln as indicated for 24 h. mRNA expression was measured by the standard curve method and normalized to *Hprt1*; y-axis depicts the transcript levels relative to gln+. Data are mean ± s.d. of $n = 6$ independent cultures by multiple unpaired t-test. (G) Schematic of experimental setup: 10-week-old C67Bl/6J mice were fed a 1.8% glutamine-containing or glutamine-free diet, and intraperitoneally injected with 0.9% saline or 20 mg/kg MSX in a 48 h interval, respectively. Mice were sacrificed after 7 days. (H) Total glutamine in brain tissue of mice treated as in (G). mRNA levels in isolated primary hepatocytes that were 24h-starved of gln and treated with 500 mM methionine sulfoximine (MSX) ± 2 mM of gln as indicated for 24 h. mRNA expression was measured by the standard curve method and normalized to *Hprt1*; y-axis depicts the transcript levels relative to gln+. Outliers were removed using Rout and Grubbs's test. Data are mean ± s.d. of $n = 12$ by unpaired t-test. (I) Transcript levels of SREBP2 targets in brain tissue of mice treated as in (G). mRNA expression was measured by the standard curve method and normalized to *Hprt1*; y-axis depicts the transcript levels relative to NaCl. Data are mean ± s.d. of $n = 6$ mice by multiple unpaired t-tests. Source data are available online for this figure.

## Glutamine is required for the transport-dependent proteolytic activation of SREBP2

The global decrease in mRNAs of cholesterol synthesis enzymes hinted at impaired activation of the ER-membrane bound transcription factor SREBP2 that is required for their induction (Brown and Goldstein, 1997). For activation, SREBP2 must traffic from the ER to the Golgi, where it is proteolytically cleaved by S1P and S2P and subsequently localizes to the nucleus (Espenshade et al, 2002; Schumacher and DeBose-Boyd, 2021; Yang et al, 2002). To test whether glutamine starvation inhibited SREBP2 proteolysis, we analyzed its precursor and mature forms in cells starved of glutamine for 8 h. Glutamine starvation inhibited the processing of SREBP2 with or without αKG in U2OS cells, consistent with the loss in HMGCR and FDFT1 (Fig. 5A). We observed a similar inhibition of SREBP2 processing in HepG2s following glutamine withdrawal (Fig. 5B). To confirm that glutamine rather than ammonia was required for SREBP2 maturation, we treated HepG2 cells refed ammonia or glutamine with MSX. We found that MSX-inhibition of GLUL led to a decrease in both the precursor and mature forms of SREBP2, but had a minimal effect in glutamine-fed cells (Fig. 5B). Thus, glutamine is required for the activation of SREBP2.

To address how glutamine inhibits SREBP2 activation, we turned to SCAP, which is required for the translocation of SREBP2 from the ER to Golgi (Brown et al, 2018; Hua et al, 1996). To test if glutamine starvation impaired the trafficking of SCAP, we monitored its cytoplasmic distribution in Chinese Hamster Ovary (CHO) cells that stably express GFP-SCAP and were also treated with MSX to inhibit GLUL-induction of HMGCR (Appendix Fig. S5A) (Nohturfft et al, 2000). GFP-SCAP was readily visualized in the Golgi in >75% of the glutamine-fed cells examined. In cells deprived of glutamine ± αKG, GFP-SCAP did not traffic to the Golgi and was retained in the ER (Fig. 5C,D). If the defect in SREBP2 activity was caused by its impaired ER-to-Golgi trafficking, we expected to see SREBP2 cleavage upon enforced Golgi-to-ER retro-translocation, such as with brefeldin A (BFA) that leads to the redistribution of Golgi proteins including S1P and S2P to the ER (DeBose-Boyd et al, 1999). To this end, we treated cells cultured with or without glutamine and BFA for 8 h. In the presence of αKG —which we included to control for off-target effects of BFA on mTORC1—BFA rescued SREBP2 cleavage and HMGCR in

glutamine-starved cells to levels similar to those in glutamine-fed cells (Fig. 5E). Consistent with these results, the expression of a cDNA encoding a nuclear form of SREBP2 rescued the loss of HMGCR in cells cultured without glutamine (Fig. 5F) (Sakai et al, 1996).

To exclude the possibility that glutamine starvation impaired global ER-to-Golgi transport-dependent proteolysis, we first examined ATF6. Like SREBP2, ATF6 is activated in an S1P- and S2P-dependent manner during ER stress (Ye et al, 2000). Glutamine starvation did not impair S1P- or S2P-activity as evidenced by ATF6 cleavage following treatment with tunicamycin, which causes ER stress by inhibiting N-linked glycan biosynthesis (Appendix Fig. S5B). Next, we tested the effect of glutamine starvation on general ER-to-Golgi transport. To this end, we monitored the trafficking of the Golgi enzyme mannosidase II-mCherry (MANII) with the retention using selective hooks system (RUSH) (Boncompain et al, 2012). In RUSH, MANII is fused to a streptavidin-binding peptide and can thus be retained in the ER through a luminal 'hook' ER protein that is fused to streptavidin (Boncompain et al, 2012). Biotin releases MANII from its 'hook' in the ER, thereby enabling its trafficking to the Golgi (Boncompain et al, 2012). Following biotin addition, cells cultured in glutamine-replete or glutamine-depleted conditions similarly accumulated Golgi-ManII, arguing against a general defect in ER-to-Golgi trafficking (Appendix Fig. S5C). Thus, glutamine is required for the ER-to-Golgi trafficking of the SCAP/SREBP2-complex.

## Glutamine is required for the ER-to-Golgi trafficking of SCAP independently of INSIG

Glutamine could regulate the trafficking of the SCAP/SREBP2 complex through an effect on SREBP2 or SCAP. To distinguish between these possibilities, we asked whether glutamine deprivation affected the activity of SREBP1, which, like SREBP2, depends on SCAP for its activation. We found that the SREBP1-dependent transcripts *Fasn*, *Acc*, and *Scd1* were decreased in isolated primary hepatocytes cultured without glutamine (Fig. 5G). This finding, along with previous work showing that excess glutamine stimulates SREBP processing, suggested to us that glutamine regulated mevalonate pathway activation by exerting an effect on SCAP, rather than SREBP2 (Inoue et al, 2011). Because glutamine

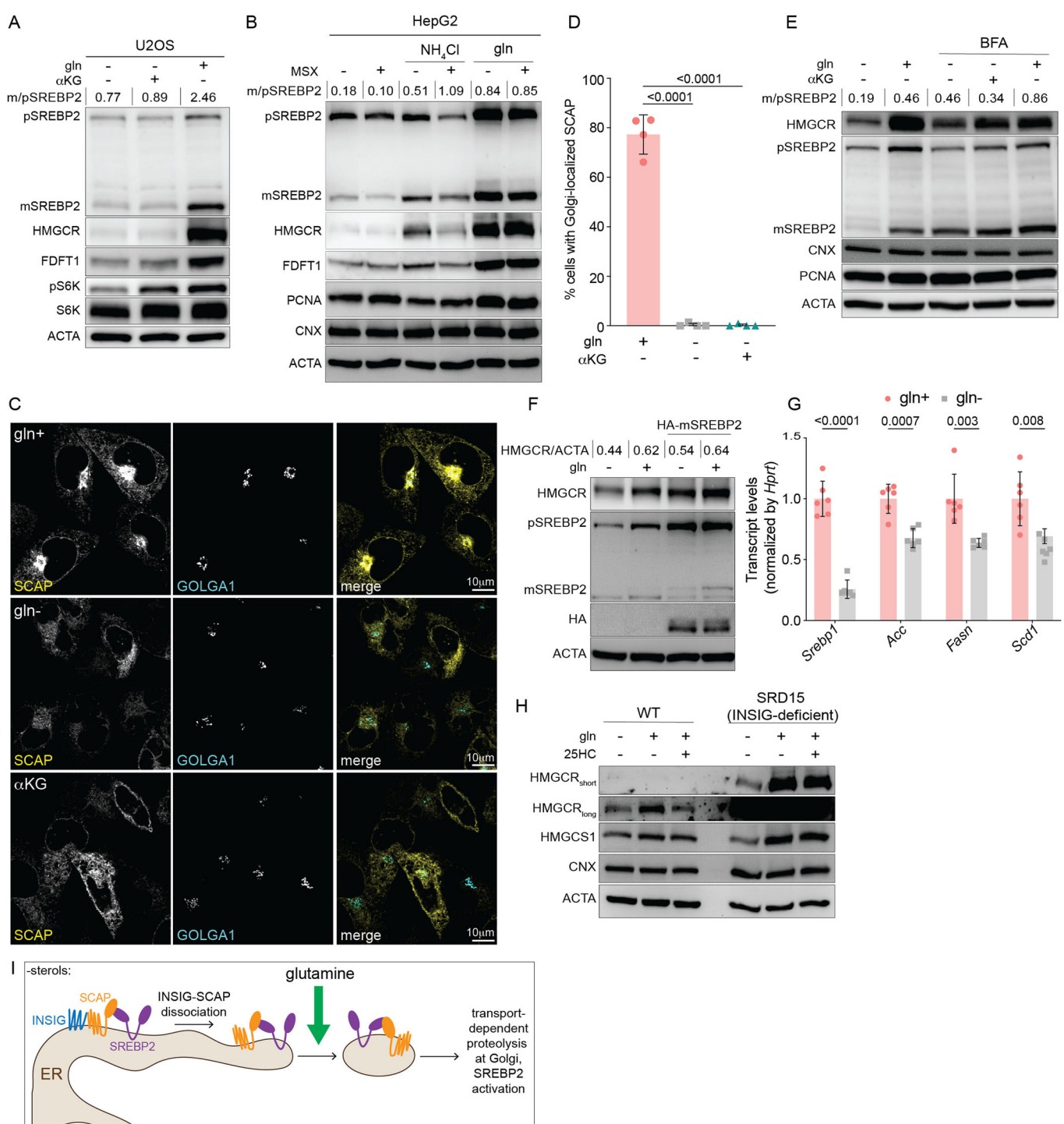

starvation did not affect total levels of SCAP, we next asked if the N-glycosylation of SCAP that enables its trafficking to the Golgi was impaired in the absence of glutamine (Cheng et al, 2016). The region of SCAP that is glycosylated faces the ER lumen (Nohturfft et al, 2000). To therefore examine N-glycosylation of SCAP, we isolated membranes from CHO cells cultured with or without glutamine. Trypsin treatment of isolated membranes yielded the expected SCAP fragments that we further digested with N-glycosidase F (PNGaseF) (Appendix Fig. S5D) (Nohturfft

et al, 2000). PNGase F treatment similarly reduced the SCAP fragment size in both glutamine-fed and glutamine-starved conditions (Appendix Fig. S5E). Thus, glutamine starvation inhibits the mevalonate pathway independently of SCAP glycosylation.

The movement of the SCAP/SREBP complex to the Golgi is gated by Insig-1 and Insig-2, ER proteins that bind SCAP in sterol-replete conditions (Brown et al, 2018). We therefore considered whether glutamine regulated the INSIG-SCAP interaction. If glutamine starvation promoted INSIG-retention of SCAP,

**Figure 5.   Glutamine is required for SCAP/SREBP2 ER-to-Golgi trafficking.**

(A) U2OS cells were cultured with w/ glutamine (gln+) or w/o gln (gln-) ± αKG for 8 h as indicated and analyzed by immunoblotting for: HMGCR, precursor (p) SREBP2, mature (m) SREBP2, FDFT1, S6K, pS6K, and actin (ACTA). (B) HepG2 cells were starved of gln for 24 h, then treated ± gln ± 10 mM NH₄Cl ± 500 mM methionine sulfoximine (MSX) as indicated for 8 h and analyzed by immunoblotting for HMGCR, SREBP2 and ACTA. (C) Representative images of eGFP-SCAP-expressing CHO cells cultured w/o gln for 24 h, and then treated with + gln of – gln ± aKG in the presence of methionine sulfoximine (MSX; 500 mM) for 24 h and processed for immunofluorescence analysis of the Golgi (GOLGA1; golgin-97). Panels show localization of eGFP-SCAP relative to the Golgi, scale bar 10 μm. (D) Percentage of cells with Golgi-localized eGFP-SCAP from experiments as in (C); data are mean ± s.d. of >100 cells counted from $n = 4$ biological replicates by one-way ANOVA. (E) HepG2s were starved of glutamine for 24 h and treated w/ brefeldin A: 0.5 mg/ml or gln for indicated time points. Samples were analyzed by immunoblotting for HMGCR, SREBP2, CNX, PCNA, and ACTA. (F) CHO cells expressing cDNA encoding for HA-tagged mature SREBP2 were cultured with or without glutamine for 8 h. Samples were analyzed by immunoblotting for HMGCR, SREBP2, HA, and ACTA. (G) SREBP1-target mRNA levels in isolated primary hepatocytes that were 24 h-starved of gln and treated with 500 mM MSX ± 2 mM of gln as indicated for 24 h. mRNA expression was measured by the standard curve method and normalized to *Hprt1*; y-axis depicts the transcript levels relative to gln+. Data are mean ± s.d. of $n = 6$ independent cultures, gln- vs. gln+ by multiple unpaired t-test. (H) Control and SRD-15 (INSIG-deficient) CHO cells were cultured ± gln ± 25HC (10 μM) for 24 h as indicated and analyzed by immunoblotting for HMGCR and ACTA. (I) Model of glutamine-based regulation of cholesterol synthesis. Glutamine was used at 2 mM and aKG at 1 mM; cells were cultured in lipid-free media for all experiments. m/p SREBP2 is the densitometry-based ratio of mature vs. precursor SREBP2. Source data are available online for this figure.

we expected that mevalonate pathway activation would be unaffected by glutamine deprivation in INSIG-deficient cells. To test this hypothesis, we turned to a previously established mutant CHO line (SRD-15) deficient in both INSIG1 and INSIG2 (Lee et al, 2005). As expected, SRD-15 cells had higher levels of HMGCR expression and were resistant to 25HC-induced degradation of HMGCR, unlike control CHO cells (Fig. 5H). By contrast, glutamine starvation drove a loss in HMGCR in both INSIG-deficient (SRD15) and WT CHO cells (Fig. 5H). These results, along with our previous finding that glutamine-starvation affects HMGCR independently of ERAD, suggest that glutamine is required for the ER-to-Golgi trafficking of SCAP in a manner independent of INSIG (Fig. 5I).

## Chronic mitochondrial dysfunction dysregulates glutamine uptake and alters cholesterol levels

Does the mevalonate pathway sense physiological levels of glutamine? To address this question, we examined HMGCR expression in glutamine-starved cells refed glutamine at a range of concentrations including 0.5 mM at which it is maintained in the bloodstream, and 2 mM, which is the concentration that is optimal for cultured cells (Eagle et al, 1956; Zhang et al, 2017). A concentration of 0.125 mM glutamine was sufficient to induce HMGCR in U2OS cells that lack GLUL activity (Fig. 6A). However, half-maximal effects of glutamine on HMGCR induction were apparent between 0.25 and 0.5 mM (Fig. 6A). This result confirmed that the mevalonate pathway was sensitive to fluctuations in glutamine levels and raised the question of whether the rewiring of glutamine metabolism that occurs during certain pathophysiological conditions impacts cholesterol metabolism.

To test this possibility, we turned to chronic disturbances of mitochondrial oxidative phosphorylation (OXPHOS) that drive increased glutamine uptake and usage to sustain the TCA cycle (Chen et al, 2018; Motori et al, 2020; Yang et al, 2014). We focused on the outer mitochondrial membrane (OMM) protein mitofusin 2 (MFN2), because its ablation impairs OXPHOS and mutations in *MFN2* in humans are associated with lipodystrophies and an axonal neuropathy known as Charcot-Marie-Tooth Disease type 2A (Capel et al, 2018; Chen et al, 2007; Pich et al, 2005; Rocha et al, 2017). U2OS cells deficient for MFN2 had decreased basal respiration, maximal respiration, ATP production and spare respiratory capacity, and consumed >2-fold more glutamine than WT cells (Fig. 6B–H). Consistent with a

defect in mitochondrial respiration, MFN2 KO cells produced more citrate m + 5 through reductive carboxylation of ¹³C₅-glutamine-derived αKG than WT cells (Fig. 6I–K) (Mullen et al, 2011). To determine whether the increased glutamine uptake in MFN2 KO cells drove changes in cholesterol metabolism, we compared mevalonate pathway activation of WT and MFN2 KO cells. Both HMGCR and the precursor and mature forms of SREBP2 were increased in MFN2 KO cells, consistent with the increase in total cholesterol (Fig. 6L–N). The addition of lipids repressed SREBP2 cleavage in both cell lines, indicating that sterol-mediated negative feedback is intact in MFN2 KO cells (Fig. 6M). Thus, the loss of MFN2 impairs OXPHOS, promotes glutamine uptake, and indirectly affects cholesterol metabolism.

If glutamine uptake drove the increase in mevalonate pathway activity in MFN2 KO cells, we expected that the difference would be attenuated in conditions of low glutamine. To test this possibility, we compared HMGCR between glutamine-starved WT and MFN2 KO cells early after glutamine refeeding. 4 h post refeeding, MFN2 KO cells had slightly higher intracellular glutamine levels although HMGCR was similar between WT and MFN2 KO cells (Fig. 6O; Appendix Fig. S6A). At later time points including 8 h and 24 h, the greater difference in glutamine levels between WT and MFN2KO cells paralleled the changes in HMGCR expression (Fig. 6O; Appendix Fig. S6A). To test whether the effect of OXPHOS deficiency on glutamine metabolism was conserved, we turned to murine embryonic fibroblasts (MEFs). Consistent with our results in human cells, we found that WT MEFs required glutamine for HMGCR expression and that *Mfn2⁻/⁻* MEFs had increased glutamine uptake and HMGCR relative to WT MEFs (Appendix Fig. S6B–D). Opposite to *Mfn2⁻/⁻* MEFs however, *Mfn1⁻/⁻* MEFs which are also deficient for mitochondrial fusion had decreased HMGCR levels relative to WT MEFs, despite that they consumed more glutamine (Appendix Fig. S6C,D). These results suggest that cholesterol homeostasis is dysregulated in a glutamine-dependent manner in a model of mitochondrial pathophysiology, but that mitochondria may communicate with the cholesterol pathways through diverse mechanisms.

## Discussion

Our data shows that glutamine controls the activation of the mevalonate pathway and raises several questions. First, why does the mevalonate pathway sense glutamine, but not glucose that is the major supplier of the carbons used to generate cholesterol?

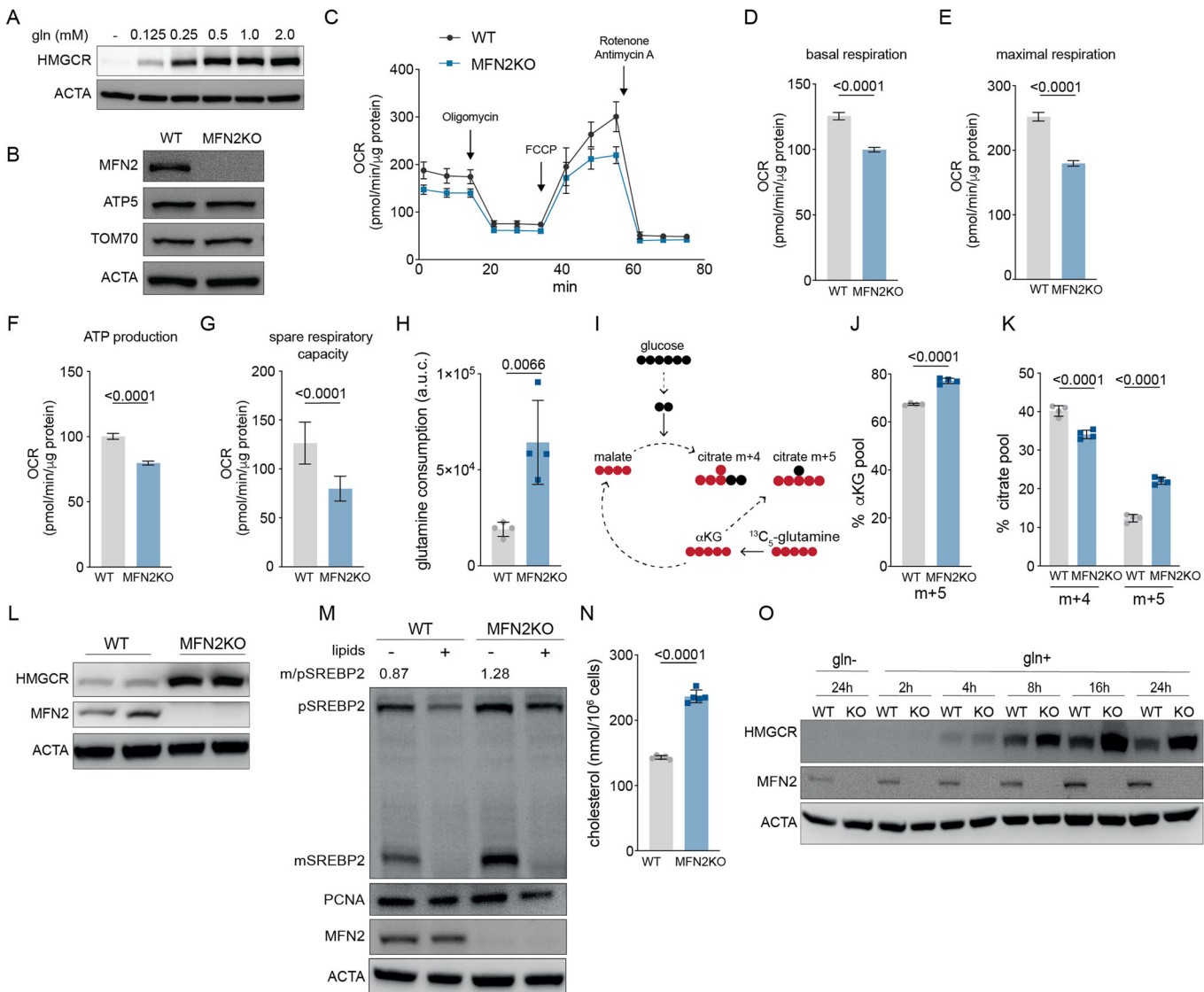

**Figure 6. Chronic mitochondrial dysfunction increases glutamine consumption and mevalonate pathway activation.**

(A) U2OS cells were cultured w/o glutamine (gln) for 24 h and treated with indicated gln concentrations for 8 h. Samples were analyzed by immunoblotting for HMGCR and actin (ACTA). (B) WT and MFN2 knockout (KO) U2OS cells were serum-starved for 24 h and analyzed by immunoblot analysis for MFN2, TOM70, ACTA, and ATP synthase F1 subunit beta (ATP5) and in parallel for (C) oxygen consumption analyses including (D) basal respiration; (E) maximal respiration; (F) ATP production; (G) spare respiratory capacity. (H) Total glutamine in DMEM versus glutamine in media 8 h after culture with WT or MFN2 KO cells. (I) During mitochondrial dysfunction glutamine feeds citrate synthesis via reductive carboxylation of αKG. (J) m + 5 αKG and (K) m + 4/5 citrate in U2OSs cultured for 24 h with 1 mM $^{13}C_5$-glutamine. Data are mean ± s.d. of $n = 4$ independent cultures by unpaired t-test (J) or (K) two-way ANOVA. (L) WT and MFN2KO U2OSs in lipid-free culture for 24 h were analyzed by immunoblotting for HMGCR, MFN2, and ACTA. (M) WT and MFN2-KO U2OSs cultured in lipid-free media overnight followed by 24 h ± lipids were analyzed by immunoblotting for SREBP2, the nuclear protein PCNA, MFN2, and ACTA; m/p is the densitometry-based ratio of mature vs. precursor SREBP2. (N) Total cholesterol levels in WT and MFN2KO U2OSs in lipid-free culture for 24 h. Data are mean ± s.d. of $n = 5$ independent cultures by unpaired t-test. (O) WT and MFN2KO U2OSs in gln-free media for 24 h were refed gln for indicated times and analyzed by immunoblotting for HMGCR, MFN2, and ACTA. Glutamine is used at 2 mM unless otherwise stated. Cells were cultured in lipid-free media for all experiments except as indicated in 6 M. Source data are available online for this figure.

Glutamine is the preferred substrate of anaplerotic reactions that sustain the TCA cycle. Cells therefore rely on glutamine to replenish TCA cycle intermediates that are extracted to fuel anabolic pathways such as lipid and heme biosynthesis (Owen et al, 2002; Yang et al, 2014). Without glutamine to sustain the TCA cycle, catabolic oxidation leads to greatly diminished concentrations of TCA cycle intermediates and restricts their use in synthetic reactions (Owen et al, 2002; Yang et al, 2014). The availability of

glutamine is therefore directly correlated with the anabolic potential of the cell. We therefore propose that communicating its levels would serve to relay precursor abundance to metabolic pathways that consume them.

Why is glutamine synthesis unique to certain cell types? Glutamine is classified as a non-essential amino acid because it is synthesized from ammonia and glutamate by GLUL. However, we found that U2OS cells that express low levels of GLUL mRNA and protein were

deficient for glutamine synthesis, unlike HepG2s that have high GLUL activity. Because glutamine is the most abundant amino acid in the human bloodstream, cellular reliance on its uptake may provide sufficient glutamine and preclude futile cycling between ATP-dependent GLUL and glutaminase, which catalyzes the breakdown of glutamine to glutamate (Zhang et al, 2017). So then why do cells from certain tissues exhibit high GLUL activity? Beyond generating glutamine, GLUL detoxifies ammonia in the liver and extrahepatic tissues and protects against glutamate excitotoxicity in the brain (Zhou et al, 2020). Our findings that GLUL sustains HMGCR expression and that dietary glutamine deprivation reduces transcripts of mevalonate pathway enzymes in brain tissue raise the additional possibility that GLUL maintains cholesterol homeostasis in tissues that rely on its de novo synthesis; the brain is unique in this regard because it contains approximately 20% of total body cholesterol but is inaccessible to peripheral lipoproteins that transport cholesterol throughout the body (Zhang and Liu, 2015). Furthermore, the GLUL promoter was recently found to contain a sterol regulatory element-binding protein 1 (SREBP1) responsive element, supporting a link between GLUL and the regulation of lipid synthesis (Jhu et al, 2021).

How do dysfunctional mitochondria communicate with the mevalonate pathway? Because mitochondria have such diverse functions ranging from supplying essential metabolites for anabolic processes to regulating cell fate and function, the mechanistic etiology of mitochondrial diseases is often unclear. We found that the metabolic rewiring that occurs during chronic mitochondrial OXPHOS dysfunction due to MFN2 loss drives glutamine uptake and increases cellular cholesterol. However, the loss of MFN1 diminished HMGCR levels, as did acute inhibition of mitochondrial translation with actinonin (Appendix Figs. S5B and S6C) (Mick et al, 2020; Richter et al, 2013; Wall et al, 2022). Thus, acute and chronic mitochondrial dysfunctions differentially dysregulate the mevalonate pathway. Reciprocally, imbalances in cholesterol metabolism can have profound effects on cellular function and organismal health, which raises the question: does dysregulated cholesterol metabolism contribute to the pathophysiology of mitochondrial dysfunction in genetic and neurodegenerative diseases, and ageing? Addressing this question may shed light on pathways amenable to therapeutic targeting in mitochondria-related diseases.

Last, how does the mevalonate pathway sense glutamine? Our results support a model in which the sensing of glutamine is required for the ER-to-Golgi trafficking of the SCAP/SREBP complex in an INSIG-independent manner. Thus, there may exist additional glutamine-regulated steps that enable the release of the SCAP/SREBP complex from the ER, or promote the interaction between SCAP and the COPII proteins that mediate its transport to the Golgi (Fig. 5I) (Sun et al, 2007).

Our study of the role of the metabolic inputs of the mevalonate pathway led to the discovery of glutamine sensing as a toggle switch for cholesterol synthesis, and thus reveal a previously uncharacterized regulatory step in the mevalonate pathway.

# Methods

## Cell culture and cell lines

HeLa adenocarcinoma cells, HepG2 hepatocellular carcinoma, and Wild-type (WT) MEFs as well as $Mfn1^{-/-}$, and $Mfn2^{-/-}$ mouse embryonic fibroblasts (MEFs) were obtained from ATCC (CCL-2,

HB-8065, CRL-2991, CRL-2992, and CRL-2993, respectively). eGFP-SCAP-expressing CHO Chinese hamster ovary cells were provided by Drs. J Goldstein and M Brown (U of Texas Southwestern Medical Center); WT and MFN2 KO U2OS cells were provided by Dr. Edward A Fon (McGill University, Montreal, Canada). HFFs were provided by the Hospital of the University of Padova, Italy. Insig-deficient SRD15 CHO cells were provided by Dr. Russell A. Debose-Boyd (University of Texas Southwestern Medical Center, Dallas, Texas). AT3 and EO771s were a gift from Dr. Ana Luisa Correia (Champalimaud Foundation, Lisbon, Portugal). All cell lines were maintained in culture in complete DMEM (cDMEM: DMEM + 10% heat-inactivated FBS + 1%P/S) at 37 °C and 5% $CO_2$ except for CHO cells which were cultured in DMEM/F12 media containing 5% heat-inactivated FBS at 8.0% $CO_2$. SRD15 CHO cells were cultured in DMEM/F12 media containing 5% heat-inactivated LDPS (lipoprotein deficient serum) + 1%P/S + 10 μM SR12813 (Sigma-Aldrich, #S4194) + 1 μM 25HC (25-Hydroxycholesterol) at 8.8% $CO_2$. For all experiments, cells were cultured without FBS (DMEM and 10% $H_2O$ for cell culture + 1%P/S) 24 h prior, as well as during glutamine supplementation/starvation experiments to promote cholesterol synthesis. Cells were tested regularly for *Mycoplasma* by PCR.

## CRISPR knockout cell lines

To generate CS and SLC25A1 CRISPR KO cell lines, U2OS cells were transduced with lentiviral particles containing the following sgRNAs cloned into the pLenti CRISPRv2 (Addgene #5296): CS guide 1F: 5′-CAA CAT GGC AAG ACG GTG GT-3′; CS guide 1R: 5′-ACC ACC GTC TTG CCA TGT TG-3′; CS guide 2F: 5′-TTT TCC AAA CCT TAC CGT GG-3′; CS guide 2R: 5′-CCA CGG TAA GGT TTG GAA AA-3′; SLC25A1 guide 1F: 5′-AGA TCT CGA TGC CAC CCG CC-3′; SLC25A1 guide 1R: 5′-GGC GGG TGG CAT CGA GAT CT-3′; SLC25A1 guide 2F: 5′-CTA CGG TTC CAT CCC CAA GG-3′; SLC25A1 guide 2R: 5′-CCT TGG GGA TGG AAC CGT AG-3′, and AAVS1 F: 5′-CAC CGG GGG CCA CTA GGG ACA GGA T-3′; AAVS1 R: 5′-AAA CAT CCT GTC CCT AGT GGC CCC C-3′. Cells were selected in 3 μg/ml puromycin, single-cell clones were and validated by immunoblotting for CS and SLC25A1.

## Puromycin incorporation assay

U2OS cells were grown in DMEM with or without glutamine or leucine (91642149, MP Biomedicals) (2 mM glutamine, 0.8 mM leucine) for 8 h. For the last 30 min of treatment, 20 μM puromycin-dihydrochloride (A11138-03, Thermo Scientific) was added to wells. Cells were harvested by scraping in lysis buffer and analyzed by immunoblot analysis for puromycin.

## 5-Ethynyl-uridine incorporation assay

U2OS cells were grown in DMEM with or without glutamine or leucine (91642149, MP Biomedicals) (2 mM glutamine, 0.8 mM leucine) for 8 h. For the last 30 min of treatment, 200 mM 5-ethynyl-uridine (5-EU, E10345, Thermo Scientific) was added to wells. As a control for inhibition of 5-EU incorporation, 4 μg/mL actinomycin D (A1410, Sigma-Aldrich) was added 10 min before to the addition of 5-EU. Cells were trypsinized and fixed in 4% formaldehyde in serum-

free DMEM without glutamine or leucine (91642149, MP Biomedicals) for 15 min at 37 °C. Fixed cells were permeabilized and stained using Click-iT Plus Alexa Fluor 647 Picolyl Azide Toolkit from Thermo Scientific (C10643) according to the manufacturer's instructions and analyzed by flow cytometry.

## Immunofluorescence assay and antibodies

For immunofluorescence (IF) analysis of eGFP-SCAP-expressing CHOs, $0.25 \times 10^4$ cells were plated in a 24-well glass-bottom sensoplate, starved of glutamine for 24 h, and incubated with methionine sulfoximine at 0.5 mM with either no glutamine, 2 mM glutamine, or 1 mM α-ketoglutarate. After 24 h, cells were fixed in 4% paraformaldehyde (fresh) in prewarmed DMEM/F12 for 20 min at 37 °C, permeabilized for 20 min at RT with 0.2% triton in 1X PBS, blocked in 3% bovine serum albumin (BSA) in 1X PBS for 30 min, and incubated in golgin-97 (CST #13192) at 1:250 in 3% BSA O/N, rinsed 3× in 1X PBS, and incubated with secondary antibody anti-rabbit Alexa Fluor Plus 594 at 1:1000 for 45 min. Following 3× rinses in 1X PBS, images were taken using an Olympus IXplore SpinSR spinning disk confocal microscope. All images were taken with a 100× objective and excitation with 488 and 561 laser lines and processed via cellSens software.

## Immunoblotting and antibodies

For all experiments, unless otherwise indicated, cells were plated in complete DMEM. The next day cell monolayers were rinsed 2× with 1X PBS and cultured in DMEM overnight, followed by treatments as indicated in text. Whole cells were harvested in chilled lysis buffer (50 mM Hepes-KOH pH 7.4, 40 mM NaCl, 2 mM EDTA, and 1% Triton X-100) containing proteases and phosphatase inhibitors (ThermoFisher Scientific A32961 and Sigma 4906837001) and Benzonase (Sigma 70746-3) following the manufacturer's instructions. Lysates were subsequently shaken at 1500 rpm at 37 °C for 15 min, centrifuged for 5 min at 1200 rpm at 4 °C, and the supernatant was transferred into a fresh tube. Protein concentration was quantified using Pierce™ BCA Protein Assay Kit and after diluted with 5X SDS added to a final of 1X SDS. 10 μg of protein was applied in an SDS-PAGE gel and transferred to PVDF membranes. Membranes were blocked with PBS-0.05% Tween 20 (PBS-T) and 5% non-fat milk for 30 min. Primary antibodies were incubated in PBS-T overnight. Following incubation, blots were washed three times in PBS-T and then incubated with horseradish peroxidase (HRP)-conjugated anti-mouse IgG (CST #7076) or anti-rabbit IgG (CST #7074) at a 1:10,000 dilution for 60 min at RT and developed using a chemiluminescence system (Pierce™ ECL Western Blotting Substrate or Pierce Super-Signal™ West Atto Ultimate Sensitivity Substrate; ThermoFisher Scientific). The following antibodies were used: Puromycin (Sigma Aldrich MABE343), MFN1/2 (Abcam ab57602), MFN2 (Abnova 157H00009927-M03J), CALR (CST #12238), ACTB (CST #4970 and Proteintech 66009-1-Ig), Golgin-97 (CST #13192S), TOMM70 (Sigma HPA048020), HMGCR (Sigma AMAB90619), FDFT1 (Sigma HPA008874), SREBP2 (BD Biosciences 557037), TUBA (Proteintech 66031-1-Ig), GLUL (CST #80636S), UB (CST # 3936S), HPRT1 (Thermo Fisher Scientific PA522281), pS6K (CST #97596S), S6K (CST #9202S), ATF4 (CST # 11815S), PCNA (BD Biosciences 610664), ATF6 (CST #65880), CNX (GeneTex GTX109669 and Proteintech 10427-2-AP) and ATP5 (Thermo Fisher Scientific A21351).

## Metabolomics and Isotope labeling sample preparation

$2 \times 10^5$ cells were plated in complete DMEM in 6-well plates. The following day, cells were rinsed 2× with 1X PBS and cultured in DMEM. Treatments were started the following day as indicated in the text in 6-well plates containing. For glucose-related isotope-labeling experiments, D-Glucose-$^{13}C_6$ (Sigma #389374-1G) was administered at 25 mM in 1 mL of glutamine-free DMEM. For ammonia-related isotope-labeling experiments, ammonium label experiments Ammonium-$^{15}N$ chloride (Sigma #299251-250MG) at 10 mM. After the indicated treatments cells were washed twice with 75 mM of ammonium carbonate pH = 7.4 and the plates were frozen in liquid nitrogen and stored at −80 °C until metabolite extraction. For metabolite extraction, 60% MeOH containing internal standards buffer was added to each well of the frozen plate. The cells were then scraped, transferred to a new tube containing MTBE and EquiSPLASH™ LIPIDOMIX® (Avanti 330731-1EA), and incubated for 30 min at 1500 rpm at 4 °C. The samples were centrifuged for 10 min at $21,000 \times g$ at 4 °C. The supernatants were then collected in a new tube with LCMS-grade $H_2O$, incubated for 10 min at 1500 rpm at 15 °C, and further centrifuged for 5 min at $16,000 \times g$ at 15 °C to obtain a clear phase separation. The upper phase (apolar metabolites) and the down phase (polar metabolites) were collected in different tubes. After, the metabolite extracts were dried down in speed vac and stored at −80 °C until further analysis as described further.

## Semi-targeted liquid chromatography-high-resolution mass spectrometry-based (LC-HRS-MS) analysis of amine-containing metabolites

The LC-HRMS analysis of amine-containing compounds was performed using a QE-Plus high-resolution mass spectrometer coupled to a Vanquish UHPLC chromatography system (Thermo Fisher Scientific). In brief: 50 μL of the available 150 μL of the above-mentioned (AEX-MS method) polar phase were mixed with 25 μl of 100 mM sodium carbonate (Sigma), followed by the addition of 25 μl 2% [v/v] benzoylchloride (Sigma) in acetonitrile (UPC/MS-grade, Biosove, Valkenswaard, Netherlands). The derivatized samples were thoroughly mixed and kept at a temperature of 20 °C until analysis. For the LC-HRMS analysis, 1 μl of the derivatized sample was injected into a $100 \times 2.1$ mm HSS T3 UPLC column (Waters). The flow rate was set to 400 μL/min using a binary buffer system consisting of buffer A (10 mM ammonium formate (Sigma), 0.15% [v/v] formic acid (Sigma) in UPC-MS-grade water (Biosove, Valkenswaard, Netherlands). Buffer B consisted of acetonitrile (IPC-MS grade, Biosove, Valkenswaard, Netherlands). The column temperature was set to 40 °C, while the LC gradient was: 0% B at 0 min, 0–15% B 0–4.1 min; 15–7% B 4.1–4.5 min; 17–55% B 4.5–11 min; 55–70% B 11–11.5 min, 70–100% B 11.5–13 min; B 100% 13–14 min; 100-0% B 14–14.1 min; 0% B 14.1–19 min; 0% B. The mass spectrometer (Q-Exactive Plus) was operating in positive ionization mode recording the mass range *m/z* 100–1000. The heated ESI source settings of the mass spectrometer were: Spray voltage 3.5 kV, capillary temperature 300 °C, sheath gas flow 60 AU, aux gas flow 20 AU at 330 °C, and the sweep gas was set to 2 AU. The RF-lens was set to a value of 60. The LC-MS data analysis was performed using the TraceFinder software (Version 5.1, Thermo Fisher

Scientific). The identity of each compound was validated by authentic reference compounds, which were measured at the beginning and the end of the sequence. For data analysis the area of the protonated [M + nBz + H]+ (nBz stands for the number of benzoyl moieties attached to each compound) isotopologue mass peaks of every required compound were extracted and integrated using a mass accuracy <3 ppm and a retention time (RT) tolerance of <0.05 min as compared to the independently measured reference compounds. If no independent 12C experiments were carried out, where the pool size is determined from the obtained peak area of the 12C monoisotopologue, the pool size determination was carried out by summing up the peak areas of all detectable isotopologues per compound. These areas were then normalized, as performed for un-traced 12C experiments, to the internal standards, which were added to the extraction buffer, followed by a normalization to the protein content or the cell number of the analyzed samples. The relative isotope distribution of each isotopologue was calculated from the proportion of the peak area of each isotopologue towards the sum of all detectable isotopologues. The 13C enrichment, namely the area attributed to 13C molecules traced in the detected isotopologues, was calculated by multiplying the peak area of each isotopologue with the proportion of the 13C and the 12C carbon number for the corresponding isotopologue (the 12C and 13C monoisotopologue areas were multiplied with 0 and 1, respectively). The obtained 13C area of each isotopologue are summed up, providing the peak area fraction associated to 13C atoms in the compound. Dividing this absolute 13C area by the summed area of all isotopologues provides the relative 13C enrichment factor.

## Anion-exchange chromatography mass spectrometry (AEX-MS) for the analysis of anionic metabolites

Extracted metabolites were re-suspended in 150 µl of UPLC/MS grade water (Biosolve), of which 100 µl were transferred to polypropylene autosampler vials (Chromatography Accessories Trott, Germany) before AEX-MS analysis. The samples were analyzed using a Dionex ionchromatography system (Integrion Thermo Fisher Scientific) as described previously. In brief, 5 µL of the resuspended polar metabolite extract were injected in push-partial mode, using an overfill factor of 1, onto a Dionex IonPac AS11-HC column (2 mm × 250 mm, 4 µm particle size, Thermo Fisher Scientific) equipped with a Dionex IonPac AG11-HC guard column (2 mm × 50 mm, 4 µm, Thermo Fisher Scientific). The column temperature was held at 30 °C, while the auto sampler temperature was set to 6 °C. A potassium hydroxide gradient was generated using a potassium hydroxide cartridge (Eluent Generator, Thermo Scientific), which was supplied with deionized water (Milli-Q IQ 7000, Millipore). The metabolite separation was carried at a flow rate of 380 µL/min, applying the following gradient conditions: 0–3 min, 10 mM KOH; 3–12 min, 10–50 mM KOH; 12–19 min, 50–100 mM KOH; 19–22 min, 100 mM KOH, 22–23 min, 100–10 mM KOH. The column was re-equilibrated at 10 mM for 3 min. For the analysis of metabolic pool sizes the eluting compounds were detected in negative ion mode using full scan measurements in the mass range m/z 77–770 on a Q-Exactive HF high resolution MS (Thermo Fisher Scientific). The heated electrospray ionization (ESI) source settings of the mass spectrometer were: Spray voltage 3.2 kV, capillary temperature was set to 300 °C, sheath gas flow 50 AU, aux gas flow 20 AU at a temperature

of 330 °C and a sweep gas glow of 2 AU. The S-lens was set to a value of 60. The LC-MS data analysis was performed using the TraceFinder software (Version 5.1, Thermo Fisher Scientific). The identity of each compound was validated by authentic reference compounds, which were measured at the beginning and the end of the sequence. For data analysis the area of the deprotonated [M-H +]-1 or doubly deprotonated [M-2H]-2 isotopologues mass peaks of every required compound were extracted and integrated using a mass accuracy <3 ppm and a retention time (RT) tolerance of <0.05 min as compared to the independently measured reference compounds. If no independent 12C experiments were carried out, where the pool size is determined from the obtained peak area of the 12C monoisotopologue, the pool size determination was carried out by summing up the peak areas of all detectable isotopologues per compound. These areas were then normalized, as performed for un-traced 12C experiments, to the internal standards, which were added to the extraction buffer, followed by a normalization to the protein content or the cell number of the analyzed samples. The relative isotope distribution of each isotopologue was calculated from the proportion of the peak area of each isotopologue towards the sum of all detectable isotopologues. The 13C enrichment, namely the area attributed to 13C molecules traced in the detected isotopologues, was calculated by multiplying the peak area of each isotopologue with the proportion of the 13C and the 12C carbon number for the corresponding isotopologue (the 12C and 13C monoisotopologue areas were multiplied with 0 and 1, respectively). The obtained 13C area of each isotopologue are summed up, providing the peak area fraction associated to 13C atoms in the compound. Dividing this absolute 13C area by the summed area of all isotopologues provides the relative 13C enrichment factor. Data deposited in https://doi.org/10.5281/zenodo.13625528.

## UPLC-HRMS-based measurement of cholesterol

The UPLC-HRMS analysis of cholesterol was performed using a modified method previously (McDonald et al, 2007). In brief: The lipid fraction of the two-phase metabolite extract was re-suspended in 100 µL of 95% methanol (Optima-Grade, Thermo Fisher Scientific) and incubated at 4 °C for 15 min on a thermomixer. The re-suspended extract was centrifuged for 5 min at 16,000 × g at 4 °C and the cleared supernatant were transferred to auto-sampler vials with 200 µL glass inserts (Chromatography Accessories Trott, Germany). For the LC-HRMS analysis, 3 µL of the sample were injected into a 100 × 2.1 mm HSS T3 UPLC column (Waters). The flow rate was set to 300 µl/min using a binary buffer system consisting of buffer A (5 mM ammonium acetate (Sigma), in 85% LC-MS-grade methanol (Optima-Grade, Thermo Fisher Scientific)), while buffer B consisted of 5 mM ammonium acetate (Sigma), in 100% LC-MS-grade methanol. The column temperature was set to 40 °C, while the LC gradient was: 0 min 0% B, 2 min, 0% B 15 min; 100% B, 25 min 100% B, 25.1 min 0% B and 30 min 0% B. The mass spectrometer (Q-Exactive HF, Thermo Fisher Scientific) was operating in positive ionization mode recording the mass range m/z 250–600. The heated ESI source settings of the mass spectrometer were: Spray voltage 3.5 kV, capillary temperature 300 °C, sheath gas flow 60 AU, aux gas flow 20 AU at a temperature of 340 °C and the sweep gas to 2 AU. The S-lens was set to a value of 60 arbitrary units. Semi-targeted data analysis for the samples was performed using the TraceFinder software (Version 4.1,

Thermo Fisher Scientific). The identity of each compound was validated by authentic reference compounds, which were run before and after every sequence. Peak areas of [M-$H_2O$ + H]+ ions were extracted using a mass accuracy (<3 ppm) and a retention time tolerance of <0.05 min. Areas of the cellular pool sizes were normalized to the internal standards (Cholesterol D7 Avanti Polar Lipids), which were added to the extraction buffer, followed by a normalization to the cell number of the analyzed sample. Enrichment analysis and isotope distribution were calculated as described in the IC/Bz section. Data deposited in https://doi.org/10.5281/zenodo.13625528.

## Quantification of total cholesterol in WT and MFN2-KO cells

Cholesterol levels were determined by Liquid Chromatography coupled to Electrospray Ionization Tandem Mass Spectrometry (LC-ESI-MS/MS). 1.5 to $5 \times 10^6$ U2OS cells were homogenized in 300 µL of Milli-Q water using the Precellys 24 Homogenisator (Peqlab) at 6500 rpm for 30 s. The protein content of the homogenate was routinely determined using bicinchoninic acid. To 70 µL of homogenate 430 µL of Milli-Q water, 1.875 ml of chloroform/methanol/37% hydrochloric acid 5:10:0.15 (v/v/v), and 1.26 nmol of deuterated cholesterol-D7 (Avanti Polar Lipids) as internal standard were added. Lipids were extracted using the "One-Step Extraction" described in Özbalci et al (2013), a method modified from Bligh and Dyer (Bligh and Dyer, 1959). Dried lipid extracts were resolved in 300 µL of methanol and sonicated for 5 min. After centrifugation (12,000 × g, 5 min, 4 °C), 40 µl of the clear supernatants were transferred to autoinjector vials. LC-MS/MS analysis of cholesterol was performed as previously described (Mourier et al, 2015).

## Glutamine consumption

For U2OS cells, cells were cultured in DMEM without FBS, pelleted, and stored at −80 °C until extraction. Samples were resuspended in extraction solution (ice-cold 80% methanol), incubated for 5 min on ice, and centrifuged for 3 min, 20,000 × g, at −9 °C. Supernatant was collected and kept on dry ice. Samples were further resuspended two more times with extraction solution and treated as previously described to further collect supernatants. Samples were finally dried using a Genevac EZ2 speed vac and further processed following protocols previously described (Edwards-Hicks et al, 2020). Targeted metabolite quantification by LC-MS was carried out using an Agilent 1290 Infinity II UHPLC in line with an Agilent 6495 QQQ-MS operating in MRM mode. MRM settings were optimized separately for all compounds using pure standards. LC separation was on a Phenomenex Luna propylamine column (50 × 2 mm, 3 µm particles) as described previously (Edwards-Hicks et al, 2020). Briefly, a solvent gradient of 100% buffer B (5 mM ammonium carbonate in 90% acetonitrile) to 90% buffer A (10 mM NH4 in water) was used. Flow rate was from 1000 to 750 µL/min. The autosampler temperature was 5 C and the injection volume was 2 µL. Raw data were analyzed using the R package automRm (Eilertz, Mitterer, Büscher, in preparation). Further analysis was performed using MassHunter software (Bruker). The amount of glutamine consumed by the cells was calculated as the difference between the starting glutamine

concentration in DMEM and the glutamine concentration in DMEM after 8 h of culturing cells of the indicated genotype. The glutamine consumption was normalized to the average cell density. For MEFs, cells between $1–2 \times 10^6$ MEFs were plated in 10 cm dishes in cDMEM so as to obtain similar confluency. The next day, cells were rinsed 2× in 1X PBS and cultured in serum-free DMEM. After 24 h, 1 mL of cultured medium was analyzed by the CEDEX Bio Analyzer according to the manufacturer's instructions. Data deposited in https://doi.org/10.5281/zenodo.13625528.

## Quantitative real-time PCR

Total RNA was prepared from U2OS, AT3, EO771, and mice organs were treated as indicated in the text using TriZol (Thermo Fischer Scientific; #15596018) or RNeasy Plus Mini Kit (Qiagen 74134) according to the manufacturer's instructions. RNA was quantified by spectrophotometry using Nanodrop One (Thermo-Fisher) and purity was assessed by the absorbance ratios of 260:280 and 260:230. From an equal amount of total RNA, complementary DNA was generated with SuperScript™ VILO™ Master Mix (Thermo Fischer Scientific; # 11755050) or high-capacity RNA-to-cDNA Kit (Thermo 4387406). Real-time quantitative PCR (qPCR) was based on the SYBR green chemistry and carried out using the Applied Biosystems real-time PCR system. The following primer sequences were used for human samples: HMGCR (F: 5′-AGG CCT GTT TGC AGA TGC TAG G-3′, R: 5′-GAT GTC CTG CTG CCA ATG CT-3′), SQLE (F: 5′-TGG GCT GCT TTC TGT ATT GTC TCC-3′, R: 5′-CAC CAC TAC TGA GAA GGG CTG G-3′), FDPS (F: 5′-GCT GGT GGT TCA GTG TCT GC-3′, R: 5′-GCA AGA ACA CTG CTG GCA GAT C-3′), FDFT1 (F: 5′-GGA AAG GGC AAG CAG TGA CC-3′, R: 5′-GGA TGG TGG AGA TGA TCT GCC TT-3′), and HPRT1 (F: 5′-TGA CAC TGG CAA AAC AAT GCA-3′, R: 5′-CGT CCT TTT CAC CAG CAA GCT-3′). The following primer sequences were used for mouse samples: Srebp2 (F: 5′-ACC TAG ACC TCG CCA AAG GT-3′, R: 5′-GCA CGG ATA AGC AGG TTT GT-3′), Hmgcr (F: 5′-GCT CGT CTA CAC AAA CTC CAC G-3′, R: 5′-GCT TCA GCA GTG CTT TCT CCG T-3′), Hmgcs1 (F: 5′-GGA AAT GCC AGA CCT ACA GGT G-3′, R: 5′-TAC TCG GAG AGC ATG TCA GGC T-3′), Fdft1 (F: 5′-TCC AAA CAG GAC TGG GAC A-3′, R: 5′-AGA CGA GAA AGG CCA ATT CC-3′), Sqle (F: 5′-TGT GCG GAT GGA CTC TTC TCC-3′, R: 5′-GTT GAC CAG AAC AAG CTC CGC A-3′), Dhcr24 (F: 5′-GGT CAT GAC GGA CGA CGT A-3′, R: 5′-AGG GCT TGT AGT AAC TGC CAA T-3′), Fdps (F: 5′-GGT GGT TCA GTG TCT GCT ACG A-3′, R: 5′-GGC CCG GGA AGT CAC TGT-3′), Srebp1 (F: 5′-GGA GCC ATG GAT TGC ACA TT-3′, R: 5′-CGC CTC ATA CAG TGC TTT CAC C-3′), Acc (F: 5′-GGA GAT GTA CGC TGA CCG AG-3′, R: 5′-TAC CCG ACG CAT GGT TTT CA-3′), Fasn (F: 5′-GCT GCG GAA ACT TCA GGA AAT-3′, R: 5′-AGA GAC GTG TCA CTC CTG GAC TT-3′), Sdc1 (F: 5′-CCA AGC TGG AGT ACG TCT GG-3′, R: 5′-CAG AGC GCT GGT CAT GTA GT-3′) and Hprt1 (F: 5′-GTT GGG CTT ACC TCA CTG CT-3′, R: 5′-TCA TCG CTA ATC ACG ACG CT-3′). Ct levels were normalized according to *HPRT1* or *Hprt1* expression levels.

## RNAseq

For RNAseq analysis cells were plated and cultured as indicated in the text. Two independent experiments were performed with n = 3. RNA

was extracted, quantified, and purity was assessed as mentioned above. The RNAseq was performed for the Cologne Center for Genomics (CCG) following the protocols previously described (George et al, 2015; Peifer et al, 2015). In brief: Libraries were prepared using the Illumina® Stranded TruSeq® RNA sample preparation Kit. Library preparation started with 500 ng total RNA. After poly-A selection (using poly-T oligo-attached magnetic beads), mRNA was purified and fragmented using divalent cations under elevated temperature. The RNA fragments underwent reverse transcription using random primers. This was followed by second-strand cDNA synthesis with DNA Polymerase I and RNase H. After end repair and A-tailing, indexing adapters were ligated. The products were then purified and amplified (15 PCR cycles) to create the final cDNA libraries. After library validation and quantification (Agilent Tape Station), equimolar amounts of library were pooled. The pool was quantified by using the Peqlab KAPA Library Quantification Kit and the Applied Biosystems 7900HT Sequence Detection System. The pool was sequenced on an Illumina NovaSeq6000 sequencing instrument with a PE100 protocol. For the RNAseq analysis, Ensembl Homo sapiens release 105/hg38 was used for genomes and annotations. rRNA transcripts were removed and cDNA fasta was generated using 'gffread' (cufflinks/2.2.1) (Trapnell et al, 2012). cDNA index was build using kallisto (kallisto/0.46.1) (Bray et al, 2016). 4 million reads were pseudoaligned to reference transcriptome using kallisto/0.46.1 and RSeQC/4.0.0 used to identify mapping strand (Wang et al, 2012). A strand was identified by having more than 60% of reads mapped to it. Cases with less than 60% of reads in each strand were defined as unstranded. Reads were pseudoaligned to reference transcriptome and quantified using kallisto/0.46.1. After the removal of batch effects, differential expression was performed between "Control" and "Treatment" samples using limma/3.54.0 (Love et al, 2014). Gln+ samples were classified as Control whereas gln- samples and aKG+ samples were classified as Treatment. Pathway enrichment analysis was performed using the DAVID API (Sherman et al, 2022). Data visualization was performed with Flaski (Iqbal et al, 2021).

### RUSH assay

MannII-SBP-mCherry stable cell lines were generated via lentivirus transduction and sequential puromycin selection. A second-generation lentivirus packaging system was used. The plasmids for the lentivirus production were obtained from Addgene, psPAX2 was a gift from Didier Trono (Addgene plasmid # 12260; http://n2t.net/addgene:12260; RRID:Addgene_12260), pMD2.G was a gift from Didier Trono (Addgene plasmid # 12259; http://n2t.net/addgene:12259; RRID:Addgene_12259) and pCDH_Str-KDEL_ManII-SBP-mCherry was a gift from Franck Perez (Addgene plasmid # 65259; http://n2t.net/addgene:65259; RRID:Addgene_65259). For Lentivirus production, 10 million HEK 293T cells were seeded on a 15 cm plate, after 24 h the cells were transfected with pCDH_Str-KDEL_ManII-SBP-mCherry, psPAX2 and pMD2.G mix in a ratio 4:2:1 using $CaPO_4$ precipitation. The virus-containing medium was collected, centrifuged, and filtered with 0.2 μM filters. Two million HeLa cells were infected in a 10 cm plate on two consecutive days. Polybrene (#TR-1003-G, Sigma) at a final concentration of 10 μg/mL was added to the medium to increase the infection efficiency. After 24 h from the last infection, the cells were selected with puromycin (#P9620-10ML, Sigma) at the final concentration of 1 μg/mL. For life imaging, HeLas stably expressing pCDH_Str-KDEL_ManII-SBP-mCherry were plated on 6-well CELLview

glass-bottom cell culture dishes (Greiner Bio-One) and treated as indicated in the text and imaged at the start and following biotin addition using an Olympus IXplore SpinSR 50 mm spinning disk confocal microscope. Live cell imaging was performed in DMEM and 10% $H_2O_2$ for cell culture + 1%P/S with incubation at 37 °C and 5% $CO_2$. All images were taken with a 100X/1.35 silicon oil objective and excitation with the 561 laser lines, using ORCA-Flash4.0 cameras (Hamatsu), and cellSens Software.

### OCR measurement

Oxygen consumption rate (OCR) was measured using Seahorse XF Analyzer. A total of 32,000 cells per well were plated in XF 96 cell culture microplates in DMEM medium supplemented with 10% FBS. The next day, DMEM was replaced by Seahorse XF medium supplemented with pyruvate (1 mM), glutamine (1 mM), and either glucose or galactose (10 mM). The OCR was measured according to the manufacturer's protocol (XF Cell Mito Stress Test Kit) at basal level and after injections with oligomycin (2 μM), FCCP (0.5 μM), and a combination of antimycin A (0.5 μM) and rotenone (0.5 μM). To normalize OCR data, protein concentration per well was determined using a bicinchoninic acid assay. The data were analyzed by Seahorse Wave Desktop software.

### mSREBP2 and GLUL-myc expression

The mature form of SREBP2 that is nuclear targeted (mSREBP2; aa 1–484) was amplified from plasmid cDNA and Gateway cloned into pDONR221 with an N-terminal HA tag. pDONR221-mSREPB2 were recombined into the pLenti6.3 V5-DEST expression vector using Gateway cloning following the manufacturer's procedure (Invitrogen). Constructs were verified by sequencing. HA-SREBP2 aa. 1–484 primers: Forward primer ATTB1 (5′-GGG GAC AAG TTT GTA CAA AAA AGC AGG CTT AGC CAC CAT GTA CCC ATA CGA TGT TCC AGA TTA CGC TGA CGA CAG CGG CGA GCT GGG TGG-3′) and Reverse primer ATTB2 (5′-GGG GAC CAC TTT GTA CAA GAA AGC TGG GTT TCA CAG AAG AAT CCG TGA GCG GTC TAC CAT-3′). The cDNA for GLUL-myc expression was purchased from Origene (CAT#: MR205788).

### Mice diet and MSX injections

Male *Mus musculus* C57BL/6J mice (n = 30) at 8 weeks of age were purchased from Charles River Laboratories and housed in a temperature-controlled environment under a 12-h light-dark cycle in pathogen-specific conditions. After 2 weeks of acclimatization, mice were randomized into two groups (n = 15) for 8 days: (1) Control diet (1.8% gln; Ssniff diet, #S9159-E764) + 0.9% saline solution injections and (2) Glutamine-deficient diet (0% gln; Ssniff diet, #S9159-E762) + Glutamine-depleted diet (0% gln; Ssniff diet, #S9159-E762) + MSX (20 mg/kg diluted in saline) injections as previously described (Ghoddoussi et al, 2010). Mice were intraperitoneally injected 3X every 48 h. At day 7, mice were euthanized by anesthesia (166.7 mg/kg of ketamine and 23.8 mg/kg of Xylazine in 0.9% NaCl) injection followed by cervical dislocation. One mouse treated with the glutamine-deficient diet + MSX was euthanized before the end of the experiment due to fast weight loss (>10%). Brain and liver tissue was collected, rinsed in 1XPBS, snap-frozen in liquid nitrogen, and stored at −80 °C until subsequent analysis. Tissues were disintegrated using a TissueLyser

(Qiagen) and analyzed for total glutamine as described above. For gene expression, tissues of 6 mice were extracted with TriZol (Thermo Fischer Scientific; #15596018) according to the manufacturer's instructions.

### Animal Welfare Statement

All the described experiments received approval from the Institutional Ethical Committee on Animal Experimentation of the Amsterdam UMC and adhere to national guidelines; animal permit approval: AVD11800202317543.

### Primary hepatocyte isolation and culture

Primary hepatocyte isolation and culture were performed as previously described (Maslansky and Williams, 1982). In brief: mice were anesthetized and liver perfusion was performed first with perfusion solution and after with collagenase solution. The liver was removed from the animal, transferred to a Petri dish containing perfusion solution, and gently combed until turned into a homogenate. The liver homogenate was passed through a sieve into 50 mL tubes. Hepatocytes were washed, counted, and plated in commercial collagen-coated plates (Corning, Cat #734-0274 or 734-0295). The cells were left to attach to the dish for 3 h and the medium was replaced with DMEM 1 g/L glucose + 100 nM of Insulin + 500 nM Dexamethasone.

### 14C-Cholesterol synthesis quantification

The incorporation of 14C-Acetate into cholesterol was measured as previously described (Zelcer et al, 2014). In brief: $2 \times 10^5$ cells were plated and serum-starved overnight. The next day, cells were cultured as indicated in the presence of 0.1 µCi/ml 14C-acetate (American Radiolabeled Chemicals). After 24 h, cells were washed twice with 1XPBS, lysed in 0.1 M NaOH. The radioactive content of the non-saponifiable lipid fraction was determined through scintillation counting. Samples were normalized to the protein content as measured with BCA. Cholesterol synthesis is expressed as DPM/mg cellular protein.

### SCAP N-glycosylation

The analysis of SCAP N-glycosylation was performed as previously described (Cheng et al, 2016). In brief: $5 \times 10^6$ eGFP-SCAP-expressing CHO cells were seeded in a 10 cm dish and FBS starved overnight. The following morning, cells were treated with or without in the presence of MSX (500 µM). After 24 h, cells were washed in PBS, scraped in 1 mL of PBS, and centrifuged at 1200 rpm for 5 min at 4 °C. Cell pellets were resuspended in Buffer 1 (10 mM HEPES-KOH (pH 7.6), 10 mM KCl, 1.5 mM MgCl$_2$, 1 mM sodium ethylenediaminetetraacetic acid (EDTA), 1 mM sodium ethylene glycol tetraacetic acid (EGTA), 250 mM sucrose, and a mixture of protease inhibitors) and incubated on ice for 30 min. Cells were next passed through a 22 G × 1 1/2 needle 30 times and centrifuged at $890 \times g$ at 4 °C for 5 min to isolate nuclei. The supernatant was centrifuged at $20,000 \times g$ for 20 min at 4 °C to isolate the membrane fractions. The membrane pellets were resuspended in 114 µL of Buffer 2 (10 mM HEPES·KOH (pH 7.6), 10 mM KCl, 1.5 mM MgCl$_2$, 1 mM sodium EDTA, and 100 mM NaCl). The resuspension was divided into 2 aliquots of 57 µL. 1 µL

of trypsin (1 µg/µL) was added to each aliquot followed by incubation for 30 min at 30 °C. The reaction was stopped by adding 2 µL (400 units) of soybean trypsin inhibitor. In one of the aliquots, 10 µL of solution containing 3.5% (wt/vol) SDS and 7% (vol/vol) 2-mercaptoethanol was heated at 100 °C for 10 min. In addition, 7 µL of 0.5 M sodium phosphate (pH 7.5), 7 µL of 10% (v/v) Nonidet P-40, and 2 µL (0.5 U/µL) of PNGase F were added to the same aliquot and incubated at 37 °C for 3 h. 5X SDS loading buffer was added to a final of 1X SDS, samples were heated at 100 °C for 10 min and applied in an SDS-PAGE gel. Anti-SCAP (9D5) antibody (Santa Cruz, sc-13553) was used to detect N-glycosylation and total SCAP protein.

### Cell cycle analysis

$4 \times 10^6$ cells were plated in complete DMEM in 10 cm plates. The following day, cells were rinsed 1X PBS and cultured in DMEM overnight. Treatments were started the following day as indicated in the text. After 8 h, cells were lifted from the dishes, pelleted, and fixed for 20 min in 4% PFA on ice. After, cells were rinsed with FACs Buffer (1X PBS + 2% FBS) and incubated with FACs buffer containing 0.1% (v/v) Triton X-100 and 10 µg/mL of DAPi for 20 min at RT. The cell cycle was analyzed using LSRFortessa™ X-20, Violet (405 nm) laser and acquiring the samples at low speed. Further analyses were made using FlowJo™ Software.

### Statistical analyses

All statistical analyses were performed using one-way ANOVA, two-way ANOVA, or an unpaired t-test or multiple unpaired t-tests in GraphPad Prism 9 software and are indicated accordingly.

## Data availability

All data are available in the supplementary materials or have been deposited in https://zenodo.org/records/13625528 for metabolomics and for RNAseq https://www.ncbi.nlm.nih.gov/geo/query/acc.cgi?acc=GSE276327. The source data of this paper are collected in the following database record: biostudies:S-SCDT-10_1038-S44318-024-00269-0.

## Peer review information

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

## Acknowledgements

We thank the Cologne Center for Genomics (CCG), MPI-AGE Metabolomics Core, the MPI-AGE Bioinformatics Core in particular Ayesha Iqbal and Jorge Boucas, the CECAD lipidomics facility in particular Susanne Brodesser, and Thomas Langer (MPI-AGE) for the generous sharing of reagents. We thank Dr. Ana Luisa Correia for reagents and the flow cytometry facility at the Champalimaud Foundation. We thank all members of the Pernas laboratory and Tim Bartsch and Julian Straub for manuscript feedback. We thank all members of the Zelcer lab for their technical support. Some figures created with BioRender.com

## Author contributions

**Bruna Martins Garcia**: Conceptualization; Validation; Investigation; Visualization; Methodology; Writing—original draft; Writing—review and editing. **Philipp Melchinger**: Formal analysis; Investigation. **Tania Medeiros**: Investigation. **Sebastian Hendrix**: Investigation. **Kavan Prabhu**: Investigation. **Mauro Corrado**: Investigation. **Jenina Kingma**: Investigation. **Andrej Gorbatenko**: Resources. **Soni Deshwal**: Investigation. **Matteo Veronese**: Resources. **Luca Scorrano**: Resources. **Erika Pearce**: Resources. **Patrick Giavalisco**: Resources; Formal analysis; Investigation; Writing—review and editing. **Noam Zelcer**: Resources; Supervision; Writing—review and editing. **Lena Pernas**: Conceptualization; Supervision; Funding acquisition; Investigation; Writing—original draft; Project administration; Writing—review and editing.

Source data underlying figure panels in this paper may have individual authorship assigned. Where available, figure panel/source data authorship is listed in the following database record: biostudies:S-SCDT-10_1038-S44318-024-00269-0.

## Funding

## Disclosure and competing interests statement

BMG and LP are inventors on a European patent application (No. EP23171736.4) filed by the Max-Planck-Gesellschaft. The patent application covers the use of GLUL activators or inhibitors to modulate lipid levels. The remaining authors declare no competing interests.

