## [Peer Review File · The EMBO Journal]

Glutamine sensing licenses cholesterol synthesis

Bruna Garcia, Philipp Melchinger, Tania Medeiros, Sebastian Hendrix, Kavan Prabhu, Mauro Corrado, Jenina Kingma, Andrej Gorbatenko, Soni Deshwal, Matteo Veronese, Luca Scorrano, Erika L. Pearce, Patrick Giavalisco, Noam Zelcer, and Lena Pernas

Corresponding author(s): Lena Pernas (lfpernas@mednet.ucla.edu)

Review Timeline:

Submission Date:	26th Feb 24
Editorial Decision:	29th Mar 24
Revision Received:	18th Jul 24
Editorial Decision:	21st Aug 24
Revision Received:	11th Sep 24
Accepted:	20th Sep 24

Editor: Daniel Klimmeck

Transaction Report:

Dear Dr Pernas,

Thank you again for the submission of your manuscript (EMBOJ-2024-117092) to The EMBO Journal. As mentioned earlier, your study was assessed by three reviewers with expertise in nutrient metabolism and glutamine biology, whose comments are enclosed below.

As you will see from the experts' reports, the referees acknowledge the robustness of the analysis and potential interest and value of your findings. However, they also express major concerns, which need to be addressed thoroughly to make them supportive of publication in the EMBO Journal. In more detail, both referees #1 and #2 state that the mechanistic insights into how glutamine is directly sensed by the mevalonate pathway are not sufficiently developed and point to additional experiments required to bolster your claims. They also request ruling out a global impact of glutamine on protein translation in this context as a key experiment to enhance the study (ref#1, 3rd paragraph; ref#2, pt.3). Further, reviewer #3 is concerned about generality of your findings and asks you to consider additional cell lines and including their cell cycle status. In addition, the reviewers also raise a number of issues related to the presentation of the findings, and overall discussion of the results, that would need to be conclusively addressed to achieve the level of robustness and clarity needed for The EMBO Journal.

Given the overall interest stated and broader angle of your findings, we are able to invite you to revise your manuscript experimentally to address the referees' comments. I need to stress though that we do require strong support from the referees on a revised version of the study in order to move on to publication of the work. Given the open outcome of the revision experiments I suggest considering EMBO Reports as an alternative venue for this work.

In light of the extensive experimentation requested, I would appreciate if you could contact me during the next weeks for exchange e.g. a video call to discuss your perspective on the comments and potential plan for revisions.

Please feel free to contact me if you have any questions or need further input on the referee comments.

When submitting your revised manuscript, please carefully review the instructions below.

Please feel free to approach me any time should you have additional questions related to this.

Thank you for the opportunity to consider your work for publication.

I look forward to your revision.

Best regards,

Daniel Klimmeck

Daniel Klimmeck, PhD
Senior Editor
The EMBO Journal

Instruction for the preparation of your revised manuscript:

- 1) a .docx formatted version of the manuscript text (including legends for main figures, EV figures and tables). Please make sure that the changes are highlighted to be clearly visible.
- 2) individual production quality figure files as .eps, .tif, .jpg (one file per figure).
- 3) a .docx formatted letter INCLUDING the reviewers' reports and your detailed point-by-point response to their comments. As part of the EMBO Press transparent editorial process, the point-by-point response is part of the Review Process File (RPF), which will be published alongside your paper.

4) a complete author checklist, which you can download from our author guidelines ([https://wol-prod-cdn.literatumonline.com/pb-assets/embo-site/Author Checklist%20-%20EMBO%20J-1561436015657.xlsx](https://wol-prod-cdn.literatumonline.com/pb-assets/embo-site/Author%20Checklist%20-%20EMBO%20J-1561436015657.xlsx)). Please insert information in the checklist that is also reflected in the manuscript. The completed author checklist will also be part of the RPF.

6) It is mandatory to include a 'Data Availability' section after the Materials and Methods. Before submitting your revision, primary datasets produced in this study need to be deposited in an appropriate public database, and the accession numbers and database listed under 'Data Availability'. Please remember to provide a reviewer password if the datasets are not yet public (see <https://www.embopress.org/page/journal/14602075/authorguide#datadeposition>).

7) Our journal encourages inclusion of *data citations in the reference list* to directly cite datasets that were re-used and obtained from public databases. Data citations in the article text are distinct from normal bibliographical citations and should directly link to the database records from which the data can be accessed. In the main text, data citations are formatted as follows: "Data ref: Smith et al, 2001" or "Data ref: NCBI Sequence Read Archive PRJNA342805, 2017". In the Reference list, data citations must be labeled with "[DATASET]". A data reference must provide the database name, accession number/identifiers and a resolvable link to the landing page from which the data can be accessed at the end of the reference. Further instructions are available at .

8) At EMBO Press we ask authors to provide source data for the main and EV figures. Our source data coordinator will contact you to discuss which figure panels we would need source data for and will also provide you with helpful tips on how to upload and organize the files.

Numerical data can be provided as individual .xls or .csv files (including a tab describing the data). For 'blots' or microscopy, uncropped images should be submitted (using a zip archive or a single pdf per main figure if multiple images need to be supplied for one panel). Additional information on source data and instruction on how to label the files are available at .

9) We replaced Supplementary Information with Expanded View (EV) Figures and Tables that are collapsible/expandable online (see examples in <https://www.embopress.org/doi/10.15252/emboj.201695874>). A maximum of 5 EV Figures can be typeset. EV Figures should be cited as 'Figure EV1, Figure EV2' etc. in the text and their respective legends should be included in the main text after the legends of regular figures.

11) For data quantification: please specify the name of the statistical test used to generate error bars and P values, the number (n) of independent experiments (specify technical or biological replicates) underlying each data point and the test used to calculate p-values in each figure legend. The figure legends should contain a basic description of n, P and the test applied. Graphs must include a description of the bars and the error bars (s.d., s.e.m.).

Please remember: Digital image enhancement is acceptable practice, as long as it accurately represents the original data and conforms to community standards. If a figure has been subjected to significant electronic manipulation, this must be noted in the

figure legend or in the 'Materials and Methods' section. The editors reserve the right to request original versions of figures and the original images that were used to assemble the figure.

We realize that it is difficult to revise to a specific deadline. In the interest of protecting the conceptual advance provided by the work, we recommend a revision within 3 months (27th Jun 2024). Please discuss the revision progress ahead of this time with the editor if you require more time to complete the revisions. Use the link below to submit your revision:

Referee #1:

The manuscript "Glutamine sensing licenses cholesterol synthesis" by Garcia et al. describes experiments to examine the ability of metabolites to positively regulate the activity of the mevalonate pathway. Starting with experiments to delineate the effects of either glucose or glutamine availability on cholesterol synthesis, the authors identified glutamine starvation as an inhibitor of cholesterol biosynthesis. They go on to carry out experiments to demonstrate that glutamine regulates the activation of SREBP2, the master transcriptional activator of the mevalonate pathway. Furthermore, in a physiologic dose-dependent manner they show that glutamine induces the expression of HMGCR, the rate-limiting enzyme in the mevalonate pathway. At first, this does not seem to be a surprising finding given that previous reports have implicated mTORC1 as a positive regulator of the mevalonate pathway and glutamine is known to regulate mTORC1 activity. Furthermore, ER stress is activated by defective protein trafficking as a result of reduced glutamine-dependent protein glycosylation and ER-stress can inhibit mTORC1. In well designed and carried out experiments to investigate these possibilities as well as other possibilities, including glutamine-dependent production of other nonessential amino acids or glutamine-dependent TCA cycle anaplerosis, the authors have ruled out many of indirect mechanisms through which glutamine might influence the mevalonate pathway. Based on their findings, the authors concluded that glutamine itself directly contributes to the control of the mevalonate pathway.

To explain their findings, the authors propose that glutamine availability is being used to license cholesterol synthesis because glutamine availability is directly correlated with the anabolic potential of the cell. The authors suggest that the reason this direct effect of glutamine on cholesterol synthesis has not been detected before is because endogenous glutamine synthesis is capable of supplying sufficient glutamine in certain cell types such as HepG2 cells to render them independent of extracellular glutamine as a regulator of cholesterol biosynthesis. In contrast, they show that U2OS sarcoma cells which express low levels of GLUL mRNA and protein, are deficient in endogenous glutamine synthesis and fail to synthesize cholesterol in the absence of extracellular glutamine. While the correlations the authors draw on are clear, what is lacking in the manuscript is a mechanistic insight of how the mevalonate pathway senses glutamine. The authors suggest that glutamine is directly required for the ER-to-Golgi trafficking of the SCAP-SREBP2 and discuss several potential ways where glutamine might contribute to SREBP2 processing. However, their attempts to implicate glutamine in the various steps of SREBP processing has not been substantiated experimentally.

What the mechanistic studies do implicate is a requirement for glutamine in support of SREBP-dependent transcription and translation as the major mechanisms by which glutamine contributes to mevalonate pathway regulation. It is therefore surprising that glutamine's known roles in supporting core transcription and selective/general mRNA translation had not been investigated as a potential mechanisms that could link glutamine availability to the mevalonate pathway. In tumor cell lines, such as those the authors have investigated, glutamine is often the most limiting intracellular amino acid and is difficult to even measure intracellularly using standard metabolomic techniques. Cell lines that undertake glutaminolysis often have glutamine as the limiting acid needed to support mRNA translation. As a result, both bulk translation as well as the translation of polyglutamine-containing proteins is compromised as intracellular levels of glutamine are depleted. The failure to rule out these possibilities makes it difficult to agree with the authors that they have shown that the direct sensing of glutamine regulates cholesterol synthesis.

Major Concern: Except in Figure 1, cellular levels of glutamine are not measure in the reported experiments. In Figure 1, under conditions in which cells are grown in glutamine-deficient media, intracellular glutamine levels were found to be essentially zero in U2OS cells. It would be important to demonstrate that this level of glutamine depletion does not compromise total cellular translation or transcription. Similarly, in Figure 3a, the observed failure to translate HMGCR in glutamine-deficient medium containing all of the other nonessential amino acids lacks appropriate controls. It would be important to demonstrate that the overall translation rate of the cell is comparable to the translation rate of the same cells grown in glutamine-containing medium. Similarly, the overall transcription rate of U2OS cells grown in the presence or absence of glutamine needs to be shown to demonstrate that the transcriptional effects are specific to SREBP-dependent transcription. If experiments are included in the revised manuscript to support a direct role for glutamine in supporting HMGCR protein production that is distinct from glutamine's required role in protein synthesis and SREBP1 transcription that is distinguished from support of the synthesis of

core transcriptional components, then the manuscript will be of general interest to the readership. The ability of glutamine to specifically support SREBP1-dependent transcription and HMGCR translation would suggest a novel mechanism linking metabolites to the control of specific transcriptional and translational programs.

Referee #2:

In this manuscript, the authors presented convincing data showing that glutamine depletion leads to a drastic reduction in HMGCR protein, and inhibition of SREBP-2 proteolytic activation. They also demonstrated that this effect was owing to the lack of glutamine itself but not its metabolic derivatives. However, the mechanistic conclusion that "The loss of HMGCR was due to a block in the ER-to-Golgi trafficking of SCAP that escorts SREBP2 to the Golgi complex for proteolytic activation and was independent of INSIG that inhibits SREBP2 processing" stated in their abstract is not supported by the data presented in the manuscript.

1. The authors concluded that glutamine depletion reduced HMGCR levels by inhibiting proteolytic activation of SREBP-2, a transcription factor that drives expression of HMGCR mRNA. This conclusion was not consistent with the following data shown in the manuscript:
 - A. In Fig. 4E, glutamine depletion only reduced HMGCR mRNA by ~20%. This slight reduction in mRNA levels is not consistent with the data shown in multiple figures that this treatment caused almost complete absence of HMGCR protein.
 - B. In Fig. 5E, under the condition of glutamine depletion, treatment with BFA completely restored cleavage of SREBP-2 to the level shown in glutamine-fed cells. However, this treatment did not bring HMGCR to the level shown in cells supplemented with glutamine (Compare lanes 2 and 3). This result directly contradicts with their conclusion stated in lines 305-307 that "BFA rescued SREBP2 cleavage and HMGCR in glutamine-starved cells to levels similar to those in glutamine-fed cells".
 - C. In Fig. 5F, the authors showed that in cells expressing the cleaved matured form of SREBP-2, glutamine no longer affects expression of HMGCR protein. However, this experiment is difficult to interpret as unlike the results shown in other figures, glutamine has a marginal effect on HMGCR protein even in control cells that do not express the transgene.
2. In Fig. 5I, the authors showed that glutamine depletion reduced HMGCR protein in cells deficient in both Insig proteins. Based on this observation, they concluded that "These results indicate that glutamine is required for the ER-to-Golgi trafficking of SCAP in a manner independent of INSIG" (lane 355-356). This is certainly an overinterpretation because other than regulating SCAP trafficking, INSIGs are directly involved in ERAD of HMGCR. In order to support this conclusion, it is necessary to show that SREBP-2 cleavage and SCAP trafficking to Golgi is blocked by glutamine depletion in cells deficient in INSIGs.
3. A simple mechanism that might explain the findings in this manuscript is that glutamine depletion slows down protein translation rate, which is expected to reduce the steady state level of proteins with a short half-life such as HMGCR. This mechanism may need to be ruled out before investigating mechanisms that are more specific to HMGCR.

Referee #3:

Two known checkpoints in the regulation of cholesterol synthesis are SREBP2 and HMGCR. The authors report that the amino acid glutamine is required to activate SREBP2, and that glutamine starvation inhibited cholesterol synthesis and led to a loss of HMGCR, even in the presence of glutamine derivatives. The expression of a nuclear SREBP2 rescued HMGCR levels during glutamine starvation.

The findings in this paper, a nutrient-dependent checkpoint in cholesterol synthesis, are intriguing and compelling. The biochemistry is elegant and the results of potential relevance. However, additional clarifications/experiments are needed to fully support the conclusions as outlined below.

The authors don't mention in results or figure legends which cells are used in given experiments. Most experiments seem to have been carried out in U2OS cells. The rationale for choosing this line for the entire project is unclear: is it entirely dependent on GLUL activity? Furthermore, it is essential to reproduce the findings in at least 2 cells lines in each experiment. In addition, there are cell lines, specifically prostate cancer cells, in which synthesis/accumulation of citrate is far superior than in other cells and it would be important to validate the glutamine starvation experiments in this setting. Also, both breast and prostate lines can synthesize steroid hormones to drive growth in a cell autonomous manner. If glutamine starvation is relevant in these cells, this would have important therapeutic implications. Thus, it is important to show that these mechanisms are at play in these cells. Finally, organ derived non-transformed epithelial cells would represent a much better comparator than fibroblasts.

There are no experiments showing the effects on cell cycle, apoptosis (see Fig 4 C) or proliferation as a result of glucose or glutamine starvation. Many/some of the findings (e.g. oxidative phosphorylation) may in fact depend on the state of the cell line under these conditions. In keeping with the potential biological consequence of glutamine starvation, was the reduction in HMGCR mRNA in the context of a more diffuse decrease in transcription (Fig 4 C seems to suggest this to some extent) or was it purely a targeted consequence?

Glutamine free diet is a difficult goal to achieve in vivo although the data in brain tissue are compelling. Objective circulating

biomarkers would be important to determine glutamine availability in these dietary conditions, particularly in cancer models. In general, separating the effects of glutamine starvation in normal and cancer tissue, ideally in the same organ, would be of great importance.

Dear Daniel,

Thank you for the helpful comments from the reviewers. We have addressed them experimentally in the revised manuscript. I hope the reviewers will agree that this is a complete and exciting story. I look forward to hearing from you in due course.

[Reviewers comments are in bold; our responses are in normal font]

Referee #1:

The manuscript "Glutamine sensing licenses cholesterol synthesis" by Garcia et al. describes experiments to examine the ability of metabolites to positively regulate the activity of the mevalonate pathway. Starting with experiments to delineate the effects of either glucose or glutamine availability on cholesterol synthesis, the authors identified glutamine starvation as an inhibitor of cholesterol biosynthesis. They go on to carry out experiments to demonstrate that glutamine regulates the activation of SREBP2, the master transcriptional activator of the mevalonate pathway. Furthermore, in a physiologic dose-dependent manner they show that glutamine induces the expression of HMGCR, the rate-limiting enzyme in the mevalonate pathway. At first, this does not seem to be a surprising finding given that previous reports have implicated mTORC1 as a positive regulator of the mevalonate pathway and glutamine is known to regulate mTORC1 activity. Furthermore, ER stress is activated by defective protein trafficking as a result of reduced glutamine-dependent protein glycosylation and ER-stress can inhibit mTORC1. In well designed and carried out experiments to investigate these possibilities as well as other possibilities, including glutamine-dependent production of other nonessential amino acids or glutamine-dependent TCA cycle anaplerosis, the authors have ruled out many of indirect mechanisms through which glutamine might influence the mevalonate pathway. Based on their findings, the authors concluded that glutamine itself directly contributes to the control of the mevalonate pathway.

To explain their findings, the authors propose that glutamine availability is being used to license cholesterol synthesis because glutamine availability is directly correlated with the anabolic potential of the cell. The authors suggest that the reason this direct effect of glutamine on cholesterol synthesis has not been detected before is because endogenous glutamine synthesis is capable of supplying sufficient glutamine in certain cell types such as HepG2 cells to render them independent of extracellular glutamine as a regulator of cholesterol biosynthesis. In contrast, they show that U2OS sarcoma cells which express low levels of GLUL mRNA and protein, are deficient in endogenous glutamine synthesis and fail to synthesize cholesterol in the absence of extracellular glutamine.

While the correlations the authors draw on are clear, what is lacking in the manuscript is a mechanistic insight of how the mevalonate pathway senses glutamine. The authors suggest that glutamine is directly required for the ER-to-Golgi trafficking of the SCAP-SREBP2 and discuss several potential ways where glutamine might contribute to SREBP2 processing. However, their attempts to implicate glutamine in the various steps of SREBP processing has not been substantiated experimentally.

What the mechanistic studies do implicate is a requirement for glutamine in support of SREBP-dependent transcription and translation as the major mechanisms by which glutamine contributes to mevalonate pathway regulation. It is therefore surprising that glutamine's known roles in supporting core transcription and selective/general mRNA translation had not been investigated as a potential mechanisms that could link glutamine availability to the mevalonate pathway. In tumor cell lines, such as those the authors have investigated, glutamine is often the most limiting intracellular amino acid and is difficult to even measure intracellularly using standard metabolomic techniques. Cell lines that undertake glutaminolysis often have glutamine as the limiting acid needed to support mRNA translation. As a result, both bulk translation as well as the translation of polyglutamine-containing proteins is compromised as intracellular levels of glutamine are depleted. The failure to rule out these possibilities makes it difficult to agree with the

authors that they have shown that the direct sensing of glutamine regulates cholesterol synthesis.

We thank referee #1 for their thoughtful summary of our work and feedback.

Major Concern: Except in Figure 1, cellular levels of glutamine are not measure in the reported experiments. In Figure 1, under conditions in which cells are grown in glutamine-deficient media, intracellular glutamine levels were found to be essentially zero in U2OS cells. It would be important to demonstrate that this level of glutamine depletion does not compromise total cellular translation or transcription. Similarly, in Figure 3a, the observed failure to translate HMGCR in glutamine-deficient medium containing all of the other nonessential amino acids lacks appropriate controls. It would be important to demonstrate that the overall translation rate of the cell is comparable to the translation rate of the same cells grown in glutamine-containing medium.

We thank referee #1 for this suggestion. In our revised manuscript, we address this concern by using a puromycin incorporation assay—the same strategy used by Pavlova et al., eLife, 2020, and that revealed that 48h of glutamine slows bulk translation. We show that glutamine depletion for 8h does not compromise total cellular translation, a time point at which we do see a clear loss in HMGCR (Fig. 4D). Thus, the translation of HMGCR is inhibited by glutamine-starvation, which does not compromise total cellular translation.

The goal of Fig. 3A was to determine whether ER associated degradation mediated the loss of HMGCR during glutamine starvation. We controlled for the effect of proteasome (and thus ERAD) inhibition by also probing for ubiquitin. We show that despite proteasomal inhibition, HMGCR levels are not rescued to levels of +glutamine media. We therefore conclude from this experiment that ERAD does not facilitate the loss of HMGCR.

Similarly, the overall transcription rate of U2OS cells grown in the presence or absence of glutamine needs to be shown to demonstrate that the transcriptional effects are specific to SREBP-dependent transcription. If experiments are included in the revised manuscript to support a direct role for glutamine in supporting HMGCR protein production that is distinct from glutamine's required role in protein synthesis and SREBP1 transcription that is distinguished from support of the synthesis of core transcriptional components, then the manuscript will be of general interest to the readership. The ability of glutamine to specifically support SREBP1-dependent transcription and HMGCR translation would suggest a novel mechanism linking metabolites to the control of specific transcriptional and translational programs.

We thank referee #1 for this suggestion. In our initial submission, our RNAseq analysis of U2OS cells cultured w/ or w/out glutamine shows that while the expression of mevalonate pathway genes is repressed, a subset of genes are induced during glutamine starvation — suggesting that total transcription is not inhibited (Fig. 4D-E in revised manuscript).

In our revised manuscript, we show that glutamine depletion does not compromise total cellular transcription at 8h, in which we see already see a loss in HMGCR (Fig. 4B). We therefore agree with referee #1 that our data suggest a novel mechanism linking glutamine to the transcriptional control of the mevalonate pathway.

Referee #2:

In this manuscript, the authors presented convincing data showing that glutamine depletion leads to a drastic reduction in HMGCR protein, and inhibition of SREBP-2 proteolytic activation. They also demonstrated that this effect was owing to the lack of glutamine itself but not its metabolic derivatives. However, the mechanistic conclusion that "The loss of HMGCR was due to a block in the ER-to-Golgi trafficking of SCAP that escorts SREBP2 to the Golgi complex for proteolytic activation and was independent of INSIG that inhibits SREBP2 processing" stated in their abstract is not supported by the data presented in the manuscript.

We thank referee #2 for this summary of our work.

1. The authors concluded that glutamine depletion reduced HMGCR levels by inhibiting proteolytic activation of SREBP-2, a transcription factor that drives expression of HMGCR mRNA. This conclusion was not consistent with the following data shown in the manuscript:

A. In Fig. 4E, glutamine depletion only reduced HMGCR mRNA by ~20%. This slight reduction in mRNA levels is not consistent with the data shown in multiple figures that this treatment caused almost complete absence of HMGCR protein.

We thank referee #2 for raising this point. In our submitted manuscript, we showed that glutamine depletion leads to an ~80% decrease in HMGCR mRNA in U2OS cells that are deficient for GLUL. In murine hepatocytes that exhibit high levels of GLUL expression (see tabula muris data below; <https://tabula-muris.ds.czbiohub.org/>), glutamine starvation lead to only a ~20% reduction in mRNA (Supp. Fig. 4A; Fig. 4E). The difference in reduction we ascribe to incomplete in glutamine synthase inhibition; indeed the addition of MSX does not fully repress HMGCR expression in isolated murine hepatocytes nor HpG2 cells (Fig. 2K-L)

B. In Fig. 5E, under the condition of glutamine depletion, treatment with BFA completely restored cleavage of SREBP-2 to the level shown in glutamine-fed cells. However, this treatment did not bring HMGCR to the level shown in cells supplemented with glutamine (Compare lanes 2 and 3). This result directly contradicts with their conclusion stated in lines 305-307 that "BFA rescued SREBP2 cleavage and HMGCR in glutamine-starved cells to levels similar to those in glutamine-fed cells".

We thank referee #2 for raising this point and apologize for any lack of clarity. BFA treatment inhibits mTORC1 (PMID: 25567907), and thus leads to decreases in global translation. Indeed, a comparison of lane 2 (+ gln) and lane 5 (+ gln +BFA) reveals that BFA, even in the presence of glutamine, decreases HMGCR levels. To address this, we also included BFA treatment with aKG (which restored S6K-phosphorylation during glutamine-starvation (Fig. 3B) in lane 4. These data show that BFA treatment, in the presence of aKG, rescues SREBP2 processing and HMGCR levels. We have also included a clarification in the text (lines 315-318): "In the presence of aKG—which we included to control for off-target effects of BFA on mTORC1—BFA rescued SREBP2 cleavage and HMGCR in glutamine-starved cells to levels similar to those in glutamine-fed cells (Fig. 5E)."

C. In Fig. 5F, the authors showed that in cells expressing the cleaved matured form of SREBP-2, glutamine no longer affects expression of HMGCR protein. However, this experiment is difficult to interpret as unlike the results shown in other figures, glutamine has a marginal effect on HMGCR protein even in control cells that do not express the transgene.

We thank referee #2 for raising this point— we also realized we erred in our initial description; this experiment was performed in CHO cells following 8h of glutamine starvation. In our revised manuscript we have included the ratio of HMGCR/ACTA w/ and w/out the HA-mSREB2 transgene, which reveals that expression of HA-mSREBP2 leads to a 22% increase in HMGCR during glutamine starvation.

2. In Fig. 5I, the authors showed that glutamine depletion reduced HMGCR protein in cells deficient in both Insig proteins. Based on this observation, they concluded that "These results indicate that glutamine is required for the ER-to-Golgi trafficking of SCAP in a manner independent of INSIG" (lane 355-356). This is certainly an overinterpretation because other than regulating SCAP trafficking, INSIGs are directly involved in ERAD of HMGCR. In order to support this conclusion, it is necessary to show that SREBP-2 cleavage and SCAP trafficking to Golgi is blocked by glutamine depletion in cells deficient in INSIGs.

We thank referee #2 for raising this point. In Fig. 3A, we show that glutamine-starvation affects HMGCR levels in a proteasome-independent, and thus ERAD-independent manner. In Fig. 5, we pinpoint the inhibition of the mevalonate pathway—and thus HMGCR—to a block in the transport-dependent proteolysis of SREBP2. We therefore believe our data do indeed support our conclusion. For further clarification, we have restated our conclusion to: "These results, along with our previous finding that glutamine-starvation affects HMGCR independently of ERAD, suggest that glutamine is required for the ER-to-Golgi trafficking of SCAP in a manner independent of INSIG (Fig. 5I)." (lines 365-367).

3. A simple mechanism that might explain the findings in this manuscript is that glutamine depletion slows down protein translation rate, which is expected to reduce the steady state level of proteins with a short half-life such as HMGCR. This mechanism may need to be ruled out before investigating mechanisms that are more specific to HMGCR.

We thank referee #2 for bringing up this point (which was also alluded to in referee #1's comments). In our revised manuscript, we show that glutamine-starvation at 8 hours (a time at which we already see a loss in HMGCR), does not affect global protein translation (Fig. 3D).

Referee #3:

Two known checkpoints in the regulation of cholesterol synthesis are SREBP2 and HMGCR. The authors report that the amino acid glutamine is required to activate SREBP2, and that glutamine starvation inhibited cholesterol synthesis and led to a loss of HMGCR, even in the presence of glutamine derivatives. The expression of a nuclear SREBP2 rescued HMGCR levels during glutamine starvation.

The findings in this paper, a nutrient-dependent checkpoint in cholesterol synthesis, are intriguing and compelling. The biochemistry is elegant and the results of potential relevance. However, additional clarifications/experiments are needed to fully support the conclusions as outlined below.

We thank referee #3 for this positive summary of our manuscript.

The authors don't mention in results or figure legends which cells are used in given experiments. Most experiments seem to have been carried out in U2OS cells. The rationale for choosing this line for the entire project is unclear: is it entirely dependent on

GLUL activity? Furthermore, it is essential to reproduce the findings in at least 2 cells lines in each experiment.

We thank referee #3 for raising this point. Because U2OS cells have undetectable GLUL expression, they are the perfect model to examine the effects of glutamine-starvation. That being said, we also assess several different cell lines, including primary human foreskin fibroblasts, HeLas, MEFs (murine), CHO cells (hamster) and HepG2s in which we inhibit GLUL activity. In all cell lines examined, glutamine starvation inhibits the mevalonate pathway.

In addition, there are cell lines, specifically prostate cancer cells, in which synthesis/accumulation of citrate is far superior than in other cells and it would be important to validate the glutamine starvation experiments in this setting.

We thank referee #3 for raising this point. To address question, in our submitted manuscript we had generated citrate synthase knockouts and show that depletion of citrate does not lead to a reduction of HMGCR levels (Supp. Fig. 3). If citrate is required for SREBP2 activation, than its depletion would lead to a loss of HMGCR, which it does not (Supp. Fig. 3) We also show that the addition of the anaplerotic glutamine-derivative aKG, which rescues citrate flux, does not rescue cholesterol synthesis (Fig. 1).

Also, both breast and prostate lines can synthesize steroid hormones to drive growth in a cell autonomous manner. If glutamine starvation is relevant in these cells, this would have important therapeutic implications. Thus, it is important to show that these mechanisms are at play in these cells. Finally, organ derived non-transformed epithelial cells would represent a much better comparator than fibroblasts.

We agree with referee #3 that assessing a role for glutamine-starvation on steroid synthesis would be extremely interesting—we do believe this is outside of the scope of our current claims.

However in our revised manuscript, we also show that glutamine is required for the expression of mevalonate pathway genes in the AT3 cell line (derived from primary murine mammary tumor cells) and the EO771 cell line (derived from spontaneous breast cancer of C57BL/6 mice) (Supp. Fig. 4B).

Regarding the latter point: we do indeed have experiments showing that glutamine-starvation leads to a loss of HMGCR in both non-transformed human foreskin fibroblasts (Fig. 2C), as well as liver derived non-transformed hepatocytes (Fig. 2L).

There are no experiments showing the effects on cell cycle, apoptosis (see Fig 4 C) or proliferation as a result of glucose or glutamine starvation. Many/some of the findings (e.g. oxidative phosphorylation) may in fact depend on the state of the cell line under these conditions. In keeping with the potential biological consequence of glutamine starvation, was the reduction in HMGCR mRNA in the context of a more diffuse decrease in transcription (Fig 4 C seems to suggest this to some extent) or was it purely a targeted consequence?

We thank reviewer #3 for raising this point. All of the experiments in the paper (except where indicated), were performed in serum-free media to ensure lipid-free conditions and thus promote cholesterol synthesis. In our revised manuscript, we show that glutamine starvation in these conditions does not lead to significant changes in cell cycle profiles relative to glutamine-fed conditions (Supp. Fig. 1A-B).

Regarding a global decrease in transcription explaining the effect of glutamine-starvation on the mevalonate pathway (a point also raised by referee #1): in our initial submission, our RNAseq

analysis of U2OS cells cultured w/ or w/out glutamine shows that while the expression of mevalonate pathway genes is repressed, a subset of genes are induced during glutamine starvation — suggesting that total transcription is not inhibited (Fig. 4B-C).

In our revised manuscript (and in line with a concern of referee #1 as well), we show that glutamine depletion does not compromise total cellular transcription at a time point in which we see already see a loss in HMGCR (Fig. 4B).

Glutamine free diet is a difficult goal to achieve in vivo although the data in brain tissue are compelling. Objective circulating biomarkers would be important to determine glutamine availability in these dietary conditions, particularly in cancer models. In general, separating the effects of glutamine starvation in normal and cancer tissue, ideally in the same organ, would be of great importance.

We agree with referee #3 that a glutamine-free diet is difficult to achieve in vivo, especially in highly metabolic organs such as the liver which likely generate glutamine through several routes (i.e. autophagy).

We believe that the best biomarker for glutamine is glutamine. To this end, we compared mevalonate pathway activity in an organ in which we observed a decrease in glutamine (brain) versus no decrease (liver) (Fig. 4H). Our results show that a decrease in glutamine is correlated with a decrease in mevalonate pathway activity (Fig. 4H-I). In liver tissue in which we did not see a decrease in glutamine, we also did not see a decrease in expression of mevalonate pathway genes (Supp. Fig. 4D-E).

Dear Dr Pernas,

Thank you for submitting your revised manuscript (EMBOJ-2024-117092R) to The EMBO Journal, as well as for your patience with our response at this time of the year. Your amended study was sent back to the referees for their re-evaluation, and we have received comments from all of them, which I enclose below. As you will see, the experts stated that the work has been substantially improved by the revisions and they are now in favour of publication, pending minor revision.

Thus, we are pleased to inform you that your manuscript has been accepted in principle for publication in The EMBO Journal.

Please consider the remaining issues of referee #2 carefully and amend the manuscript accordingly by complementary data, or alternatively revisiting the discussion of the results.

We also now need you to take care of a number of minor issues related to formatting and data presentation as detailed below, which should be addressed at re-submission.

Please contact me at any time if you have additional questions related to below points.

As you might have seen on our web page, every paper at the EMBO Journal now includes a 'Synopsis', displayed on the html and freely accessible to all readers. The synopsis includes a 'model' figure as well as 2-5 one-short-sentence bullet points that summarize the article. I would appreciate if you could provide this figure and the bullet points.

Thank you for giving us the chance to consider your manuscript for The EMBO Journal. I look forward to your final revision.

Again, please contact me at any time if you need any help or have further questions.

Best regards,

Daniel Klimmeck

>> Please limit the number of keywords for your study to maximally five.

>> Author Contributions: Please remove the author contributions information from the manuscript text. Note that CRediT has replaced the traditional author contributions section as of now because it offers a systematic machine-readable author contributions format that allows for more effective research assessment. and use the free text boxes beneath each contributing author's name to add specific details on the author's contribution.

More information is available in our guide to authors.
<https://www.embopress.org/page/journal/14602075/authorguide>

>> Rename the current 'Competing Interests' section to 'Disclosure and Competing Interests Statement'.

>> Figures: the main figures should be uploaded as individual, high resolution figure files. The figure legends should be removed from the main figures and added to the manuscript text.

>> Section order should be corrected as follows: title page with complete author information, abstract, keywords, introduction, results, discussion, methods, data availability section, acknowledgements, disclosure and competing interests statement, references, main figure legends, tables, expanded figure legends.

>> The following should be removed from the manuscript: "List of Supplementary Materials paragraph".

>> References: adjust reference format to EMBO Journal format, 10 authors et al, and place References after the Discussion, before figure legends. Please integrate the current supplemental references.

>> Funding: information on funding needs to be part of Acknowledgments section; please complete in our online system: 'Howard Hughes Medical Institute and NWO ENW grant (M.22.034; GENESIS)'.

>> Move all current Supplemental Methods to the main manuscript 'Methods' section.

>> Appendix file: in, but needs to be in PDF; all references to "Supplementary" need to be updated to "Appendix"; the ToC on the title page should have all the Appendix items and their page numbers; the correct nomenclature and manuscript callouts should be "Appendix Figure S1" etc. .

>> Please provide source data for the study as to the separate request e-mail by my colleague Hannah Sonntag.

>> Callouts: please revisit callouts to Supplemental Tables S1-S3 in the running text.

>> Data availability section: please rename the current 'Data and materials availability' paragraph to 'Data availability section'; deposit the RNAseq and metabolomics data on public repositories. Annotate dataset identifiers and hyperlinks in the Data availability section.

>> Provide an animal welfare statement in the Methods section.

>> Author Checklist: complement the Ethics section with information on animal ethics. Adjust the Data availability part.

>> Consider additional changes and comments from our production team as indicated below:

Figure Legends - Comments

- Please define the annotated p values ****/*** as well as provide the exact p-values for the same in the legends of figures 5g; 6n; as appropriate.
- Please note that the exact p values are not provided in the legends of figures 1b-e, g-h; 2f-h; 4b, f, h-i; 5d; 6d-h.
- Please indicate the statistical test used for data analysis in the legends of figures 4c-d; 5g; 6n.
- Please note that information related to n is missing in the legends of figures 2g-h.
- Please note that the error bars are not defined in the legends of figures 2g-h.

Please use the link below to submit your revision:

Referee #1:

This manuscript, "Glutamine sensing licenses cholesterol synthesis" by Garcia et al has been revised to address the suggestions of the reviewers. The manuscript now includes additional data demonstrating that the glutamine-dependence of HMGCR expression is not the result of a generalized effect of glutamine-deprivation on bulk translation (revised Figure 3D), nor due to glutamine-depletion compromising total cellular transcription to a level that would lead to the reduction of HMGCR that the authors have observed (revised Figure 4B). These new data suggest that there is specificity in how glutamine influences cholesterol synthesis and further supports the authors' conclusions.

How glutamine effects ER-to-Golgi trafficking of the SCAP/SREBP Complex still remains to be determined. Nevertheless, the general readership of the journal will find the linkage between glutamine and cholesterol biosynthesis novel and intriguing. The manuscript is likely to stimulate additional investigation into how glutamine levels modulate cholesterol biosynthesis as well as investigation of the significance of this association in modulating pathophysiologic conditions associated with excess cholesterol.

Referee #2:

The revised manuscript addressed my first concern, but the rest two concerns remain unresolved.

2. An interesting finding in the manuscript is that glutamine increased HMGCR levels in an INSIG-independent manner. The authors explained this finding by concluding that glutamine stimulated ER-to-Golgi transport of SCAP in an INSIG-independent manner, and the resultant proteolytic activation of SREBP2 activated HMGCR expression. This reviewer pointed out previously that this conclusion needs experimental support by showing that glutamine stimulates SREBP2 cleavage in cells deficient in both INSIG proteins. Unfortunately, while the authors have all the reagents and experimental procedures ready for such an experiment, they did not perform this experiment in the revised manuscript. Since this is a potential ground-breaking finding, just toning down the language is not enough to address my concern under this circumstance.

3. Both this reviewer and reviewer 1 pointed out that the authors need to rule out the possibility that general protein translation was not perturbed under their glutamine-depleted experimental conditions. In the revised manuscript, they tried to demonstrate that this is indeed the case using the puromycin incorporation assay. In this assay, puromycin incorporates into elongating peptide chains under translation, causing dissociation of prematurely terminated peptides from ribosomes that can be detected through western blot. Thus, this assay measures the levels of ongoing translation that has been initiated. However, it is not suitable to determine whether the translation can be completed to produce mature proteins. Considering that glutamine depletion is more likely to affect completion of translation, this assay is not appropriate under this circumstance. The standard [³⁵S]Met pulse experiments should be performed to address this important point.

Referee #3:

I think the authors have generally answered satisfactorily the critiques. In terms of some responses however, specifically those questioning the data in Figs. 4E and 5E, the rebuttal falls short of the requests by the reviewer and does not seem entirely satisfactory. Nevertheless, I think that taken together, the data presented in the manuscript are compelling.

Dear Daniel,

Thank you for the helpful comments from the reviewers and production team. We hope we have addressed them to their and your satisfaction in our revised draft. Furthermore, we have also included a new panel that demonstrates that the expression of glutamine synthetase in U2OS cells leads to the induction of HMGCR by ammonia supplementation (confirming that glutamine synthesis is required for ammonia to induce HMGCR (Fig. 2K). We hope that you agree this is an interesting story that will bring renewed interest to the role of glutamine in licensing cholesterol synthesis, as well as the role of input sensing in metabolic regulation.

[Reviewers comments are in bold; our responses are in normal font]

Regarding formatting changes required for the revised version of the manuscript:

Please limit the number of keywords for your study to maximally five.

Done

Author Contributions: Please remove the author contributions information from the manuscript text. Note that CRediT has replaced the traditional author contributions section as of now because it offers a systematic machine-readable author contributions format that allows for more effective research assessment. and use the free text boxes beneath each contributing author's name to add specific details on the author's contribution.

Done.

Rename the current 'Competing Interests' section to 'Disclosure and Competing Interests Statement'.

Done.

Figures: the main figures should be uploaded as individual, high resolution figure files. The figure legends should be removed from the main figures and added to the manuscript text.

Done.

Section order should be corrected as follows: title page with complete author information, abstract, keywords, introduction, results, discussion, methods, data availability section, acknowledgements, disclosure and competing interests statement, references, main figure legends, tables, expanded figure legends.

Done.

The following should be removed from the manuscript: "List of Supplementary Materials paragraph".

Done.

References: adjust reference format to EMBO Journal format, 10 authors et al, and place References after the Discussion, before figure legends. Please integrate the current supplemental references.

Done.

Funding: information on funding needs to be part of Acknowledgments section; please complete in our online system: 'Howard Hughes Medical Institute and NWO ENW grant (M.22.034; GENESIS)'.

Done.

Move all current Supplemental Methods to the main manuscript 'Methods' section.

Done.

Appendix file: in, but needs to be in PDF; all references to "Supplementary" need to be updated to "Appendix"; the ToC on the title page should have all the Appendix items and their page numbers; the correct nomenclature and manuscript callouts should be "Appendix Figure S1" etc.

Done.

Please provide source data for the study as to the separate request e-mail by my colleague Hannah Sonntag.

Done.

Callouts: please revisit callouts to Supplemental Tables S1-S3 in the running text.
Revised to Table EV1-3.

Data availability section: please rename the current 'Data and materials availability' paragraph to 'Data availability section'; deposit the RNAseq and metabolomics data on public repositories. Annotate dataset identifiers and hyperlinks in the Data availability section.

Done.

Provide an animal welfare statement in the Methods section.

Done.

Author Checklist: complement the Ethics section with information on animal ethics.

Done.

Adjust the Data availability part.

Done.

Consider additional changes and comments from our production team as indicated below:

Figure Legends - Comments

Please define the annotated p values **/*****

Have indicated the exact p-values instead in all the figures

as well as provide the exact p-values for the same in the legends of figures 5g; 6n; as appropriate

Done.

Please note that the exact p values are not provided in the legends of figures 1b-e, g-h; 2f-h; 4b, f, h-i; 5d; 6d-h.

Done. We use prism for our statistical analyses, which does not provide exact values for anything less than 0.0001. Thus, we have provided exact values where we were able to.

Please indicate the statistical test used for data analysis in the legends of figures 4c-d; 5g; 6n.

Done.

Please note that information related to n is missing in the legends of figures 2g-h.

Information clarified.

Please note that the error bars are not defined in the legends of figures 2g-h.

Information clarified.

Referee #1:

This manuscript, "Glutamine sensing licenses cholesterol synthesis" by Garcia et al has been revised to address the suggestions of the reviewers. The manuscript now includes additional data demonstrating that the glutamine-dependence of HMGCR expression is not the result of a generalized effect of glutamine-deprivation on bulk translation (revised Figure 3D), nor due to glutamine-depletion compromising total cellular transcription to a level that would lead to the reduction of HMGCR that the authors have observed (revised Figure 4B). These new data suggest that there is specificity in how glutamine influences cholesterol synthesis and further supports the authors' conclusions.

How glutamine effects ER-to-Golgi trafficking of the SCAP/SREBP Complex still remains to be determined. Nevertheless, the general readership of the journal will find the linkage between glutamine and cholesterol biosynthesis novel and intriguing. The manuscript is likely to stimulate additional investigation into how glutamine levels modulate cholesterol biosynthesis as well as investigation of the significance of this association in modulating pathophysiologic conditions associated with excess cholesterol.

We thank Referee #1 for their thoughtful comments and feedback throughout the revision process.

Referee #2:

The revised manuscript addressed my first concern, but the rest two concerns remain unresolved.

2. An interesting finding in the manuscript is that glutamine increased HMGCR levels in an INSIG-independent manner. The authors explained this finding by concluding that glutamine stimulated ER-to-Golgi transport of SCAP in an INSIG-independent manner, and the resultant proteolytic activation of SREBP2 activated HMGCR expression. This reviewer pointed out previously that this conclusion needs experimental support by showing that glutamine stimulates SREBP2 cleavage in cells deficient in both INSIG proteins. Unfortunately, while the authors have all the reagents and experimental procedures ready for such an experiment, they did not perform this experiment in the revised manuscript. Since this is a potential ground-breaking finding, just toning down the language is not enough to address my concern under this circumstance.

We thank Referee #2 for their thoughtful comments and critical feedback throughout the revision process.

Previously, Referee #2 stated that: "These results indicate that glutamine is required for the ER-to-Golgi trafficking of SCAP in a manner independent of INSIG" (lane 355-356). This is certainly an overinterpretation because other than regulating SCAP trafficking, INSIGs are directly involved in ERAD of HMGCR. In order to support this conclusion, it is necessary to show that SREBP-2 cleavage and SCAP trafficking to Golgi is blocked by glutamine depletion in cells deficient in INSIGs."

We agree that the ideal experiment would have been to show SREBP-2 cleavage in cells deficient in INSIGs. Unfortunately, reagents to examine endogenous SREBP-2 cleavage in

hamster cells (INSIG-deficient cell line is SRD15 line derived from Chinese Hamster Ovary cells, are limiting. However, taken together the following results support our conclusion that glutamine is required for the ER-to-Golgi trafficking of SCAP in a manner independent of INSIG:

- glutamine depletion leads to a loss of HMGCR in a proteasome-independent manner
- glutamine depletion impairs SREBP2 cleavage in U2OS cells
- glutamine depletion impairs SREBP2 cleavage in HepG2 cells
- glutamine depletion impairs SCAP trafficking in CHO cells
- glutamine depletion leads to a loss of HMGCR in CHO cells
- glutamine depletion leads to a loss of HMGCR in INSIG-deficient CHO cells

3. Both this reviewer and reviewer 1 pointed out that the authors need to rule out the possibility that general protein translation was not perturbed under their glutamine-depleted experimental conditions. In the revised manuscript, they tried to demonstrate that this is indeed the case using the puromycin incorporation assay. In this assay, puromycin incorporates into elongating peptide chains under translation, causing dissociation of prematurely terminated peptides from ribosomes that can be detected through western blot. Thus, this assay measures the levels of ongoing translation that has been initiated. However, it is not suitable to determine whether the translation can be completed to produce mature proteins. Considering that glutamine depletion is more likely to affect completion of translation, this assay is not appropriate under this circumstance. The standard [35S]Met pulse experiments should be performed to address this important point.

We thank Referee #2 for having raised this important point previously. In our revised manuscript, we addressed this concern by using a puromycin incorporation assay. This approach was used by Pavlova et al., eLife, 2020 to investigate exactly the question raised by Referee #2. This work showed that 48h of glutamine-starvation slows bulk translation, but does not show that glutamine depletion affects the completion of translation. Using the same method, we show that glutamine depletion for 8h does not compromise total cellular translation, a time point at which we do see a clear loss in HMGCR (Fig. 4D). This result, along with our data showing that 8-24h of glutamine starvation does not impair global transcription, but does impair SREBP2 cleavage and SCAP trafficking, supports our conclusion that glutamine depletion blocks cholesterol synthesis independently of a global effect on translation.

Referee #3:

I think the authors have generally answered satisfactorily the critiques. In terms of some responses however, specifically those questioning the data in Figs. 4E and 5E, the rebuttal falls short of the requests by the reviewer and does not seem entirely satisfactory. Nevertheless, I think that taken together, the data presented in the manuscript are compelling.

We thank Referee #3 for their thoughtful comments and feedback throughout the revision process.

Dear Dr Pernas and team,

Thank you for submitting the revised version of your manuscript. I have now evaluated your amended manuscript and concluded that the remaining minor concerns have been sufficiently addressed.

I am thus pleased to inform you that your manuscript has been accepted for publication in the EMBO Journal.

Related, I kindly ask for your consent on keeping the referee figures included in this file.

On a different note, I would like to alert you that EMBO Press offers a format for a video-synopsis of work published with us, which essentially is a short, author-generated film explaining the core findings in hand drawings, and, as we believe, can be very useful to increase visibility of the work. Please see the following link for representative examples and their integration into the article web page:

<https://www.embopress.org/doi/full/10.15252/emj.2019103932>

Best regards,

Daniel Klimmeck

Daniel Klimmeck, PhD
Senior Editor
The EMBO Journal
EMBO
Postfach 1022-40
Meyerhofstrasse 1
D-69117 Heidelberg
contact@embojournal.org
Submit at: <http://emboj.msubmit.net>